# In vivo fluorescence lifetime imaging of macrophage intracellular metabolism during wound responses in zebrafish

Veronika Miskolci[1], Kelsey E Tweed[2,3], Michael R Lasarev[4], Emily C Britt[2,5], Alex J Walsh[2†], Landon J Zimmerman[1], Courtney E McDougal[1], Mark R Cronan[6‡], Jing Fan[2,5], John-Demian Sauer[1], Melissa C Skala[2,3]*, Anna Huttenlocher[1,7]*

[1]Department of Medical Microbiology and Immunology, University of Wisconsin-Madison, Madison, United States; [2]Morgridge Institute for Research, Madison, United States; [3]Department of Biomedical Engineering, University of Wisconsin-Madison, Madison, United States; [4]Department of Biostatistics & Medical Informatics, University of Wisconsin-Madison, Madison, United States; [5]Department of Nutritional Sciences, University of Wisconsin-Madison, Madison, United States; [6]Department of Molecular Genetics and Microbiology, Duke University School of Medicine, Durham, United States; [7]Department of Pediatrics, University of Wisconsin-Madison, Madison, United States

*For correspondence:
mcskala@wisc.edu (MCS);
huttenlocher@wisc.edu (AH)

Present address: †Department of Biomedical Engineering, Texas A&M University, College Station, United States; ‡In Vivo Cell Biology of Infection Unit, Max Planck Institute for Infection Biology, Berlin, Germany

Competing interest: The authors declare that no competing interests exist.

**Abstract** The function of macrophages in vitro is linked to their metabolic rewiring. However, macrophage metabolism remains poorly characterized in situ. Here, we used two-photon intensity and lifetime imaging of autofluorescent metabolic coenzymes, nicotinamide adenine dinucleotide (phosphate) (NAD(P)H) and flavin adenine dinucleotide (FAD), to assess the metabolism of macrophages in the wound microenvironment. Inhibiting glycolysis reduced NAD(P)H mean lifetime and made the intracellular redox state of macrophages more oxidized, as indicated by reduced optical redox ratio. We found that TNFα+ macrophages had lower NAD(P)H mean lifetime and were more oxidized compared to TNFα− macrophages. Both infection and thermal injury induced a macrophage population with a more oxidized redox state in wounded tissues. Kinetic analysis detected temporal changes in the optical redox ratio during tissue repair, revealing a shift toward a more reduced redox state over time. Metformin reduced TNFα+ wound macrophages, made intracellular redox state more reduced and improved tissue repair. By contrast, depletion of STAT6 increased TNFα+ wound macrophages, made redox state more oxidized and impaired regeneration. Our findings suggest that autofluorescence of NAD(P)H and FAD is sensitive to dynamic changes in intracellular metabolism in tissues and can be used to probe the temporal and spatial regulation of macrophage metabolism during tissue damage and repair.

## Editor's evaluation

Immunometabolism is an emerging field, and to understand immune cell metabolism during inflammation and infection is of great interest. In this report, cutting edge microscopy techniques and innovative zebrafish models are used to characterize the metabolism of macrophages in situ. In the future, fluorescence microscopy approaches pioneered using zebrafish may illuminate strategies to therapeutically manipulate metabolism in human immune cells.

## Introduction

Macrophages are innate immune cells that play key functions in tissue repair (*Krzyszczyk et al., 2018*; *Wynn and Vannella, 2016*). The heterogeneity and diversity of macrophage phenotypes and functions are well documented in vitro (*Martinez and Gordon, 2014*; *Mills et al., 2014*; *Murray,*

**Table 1.** Definition of autofluorescence imaging endpoints.

| Endpoints | Definition | Interpretation |
|---|---|---|
| Optical redox ratio<br>*Chance et al., 1979* | $\dfrac{I_{NAD(P)H}}{I_{NAD(P)H}+I_{FAD}}$ | Increase in optical redox ratio (ORR) = more reduced intracellular environment; likely increase in glycolysis; decrease in ORR = more oxidized intracellular environment; likely decrease in glycolysis. (I, intensity) |
| Nicotinamide adenine dinucleotide (phosphate) (NAD(P)H) | Short lifetime of NAD(P)H | Free/unbound NAD(P)H |
| NAD(P)H $\tau_2$ | Long lifetime of NAD(P)H | NAD(P)H bound to a protein |
| NAD(P)H $\alpha_1$ | Fractional component of free NAD(P)H | $\alpha_2$ is fractional component of bound NAD(P)H, $\alpha_1 + \alpha_2 = 1$; quantifies the pools of NAD(P)H in free and bound states |
| NAD(P)H mean lifetime () | $\tau_m = \tau_1 \alpha_1 + \tau_2 \alpha_2$ | Weighted average of individual lifetime endpoints (); one can look at changes in individual endpoints to see what drives changes in $\tau_m$; for instance, a decrease in $\tau_m$ can be due to increase in $\alpha_1$, decrease in $\tau_1$, and/or decrease in $\tau_2$ |
| Flavin adenine dinucleotide (FAD) $\tau_1$ | Short lifetime of FAD | FAD bound to a protein |
| FAD $\tau_2$ | Long lifetime of FAD | Free/unbound FAD |
| FAD $\alpha_1$ | Fractional component of bound FAD | $\alpha_2$ is fractional component of free FAD, $\alpha_1 + \alpha_2 = 1$; quantifies the pools of FAD in free and bound states |
| FAD $\tau_m$ | $\tau_m = \tau_1 \alpha_1 + \tau_2 \alpha_2$ | Weighted average of individual lifetime endpoints ($\tau_1, \tau_2, \alpha_1, \alpha_2$); one can look at changes in individual endpoints to see what drives changes in $\tau_m$; for instance, a decrease in $\tau_m$ can be due to increase in $\alpha_1$, decrease in $\tau_1$, and/or decrease in $\tau_2$ |
| Optical Metabolic Imaging (OMI) index<br>*Walsh and Skala, 2015* | $\dfrac{ORR_i}{<ORR>} + \dfrac{NAD(P)H\ \tau_{mi}}{<NAD(P)H\ \tau_m>} - \dfrac{FAD\ \tau_{mi}}{<FAD\ \tau_m>}$ | Composite measure of mean-centered optical redox ratio and mean lifetimes of NAD(P)H and FAD; increase in the OMI index corresponds to increased redox ratio, and increased NAD(P)H and FAD protein-binding activities |

*2017*). However, there is a gap in understanding macrophage phenotypes in interstitial tissues during tissue damage and repair. This is particularly important because distinct macrophage populations play important roles in wound healing and tissue regeneration.

Macrophages are commonly described as classically (M1) or alternatively (M2) activated, with both subsets playing critical roles in wound healing (*Krzyszczyk et al., 2018*; *Wynn and Vannella, 2016*). The M1/M2 classification, especially in the context of in vivo biology, is controversial (*Murray et al., 2014*; *Orecchioni et al., 2019*), and more likely represents a continuum of activation states. The importance of metabolic regulation of macrophage function was not appreciated until more recently, when it was recognized that some metabolic pathways are profoundly altered in classically activated macrophages (*Jha et al., 2015*; *Tannahill et al., 2013*). For example, classically activated macrophages are glycolytic, while oxidative phosphorylation is the main fuel source during alternative activation in vitro (*O'Neill et al., 2016*; *Van den Bossche et al., 2017*). This recent progress has led to the emergence of metabolic reprogramming as a hallmark of immune cell activation, and supports the premise that the metabolic state is not an outcome but rather a determinant of immune cell activation and function (*O'Neill and Pearce, 2016*; *Ryan and O'Neill, 2020*).

While it is well documented that macrophages exhibit plasticity as wounds repair and convert from M1 to M2 over the course of wound healing (*Krzyszczyk et al., 2018*), the metabolic regulation of macrophage function within interstitial tissue during wound repair remains unclear (*Caputa et al., 2019*). We need additional tools to detect the polarization and metabolic phenotypes of macrophages within interstitial tissues in live animals.

Autofluorescence imaging of the intensities and lifetimes of metabolic coenzymes is an attractive method to monitor macrophage metabolism and function in vivo. The reduced forms of nicotinamide adenine dinucleotide (phosphate) (NAD(P)H) and oxidized flavin adenine dinucleotide (FAD) are endogenous metabolic coenzymes that are autofluorescent. The fluorescence intensities of NAD(P)H and FAD can be used to determine the optical redox ratio (*Table 1*), which provides a label-free method to monitor the oxidation-reduction state of the cell (*Chance et al., 1979*). Multiple definitions of the optical redox ratio exist, but here we use NAD(P)H/(NAD(P)H + FAD), since an increase in the optical redox ratio intuitively corresponds with a more reduced intracellular environment, suggestive of an increase in glycolysis, and it normalizes the values to be between 0 and 1 (*Walsh et al., 2021*). The fluorescence lifetime measures

the time a molecule spends in the excited state before decaying back to the ground state. The fluorescence lifetimes of NAD(P)H and FAD are distinct in their free and protein-bound states, which provide a label-free measurement of their enzyme-binding activities (*Table 1*; *Georgakoudi and Quinn, 2012*; *Kolenc and Quinn, 2019*). Fluorescence lifetime imaging microscopy (FLIM) has several advantages over intensity measurements, because FLIM provides additional biological information by distinguishing the protein-bound and free states, and is not dependent on the cellular concentrations of the coenzymes (*Datta et al., 2020*; *Walsh and Skala, 2015*). Importantly, FLIM is a label-free and noninvasive method to detect metabolic changes in situ and can also resolve metabolic heterogeneity within a cell population (*Heaster et al., 2019*; *Sharick et al., 2019*; *Walsh et al., 2021*; *Walsh and Skala, 2015*).

Zebrafish represents a powerful system to study macrophage polarization and tissue repair. Live imaging has revealed the presence of both M1 (TNFα+) and M2 (TNFα−) macrophages in wounded tissues (*Miskolci et al., 2019*; *Nguyen-Chi et al., 2017*; *Nguyen-Chi et al., 2015*). Here, we performed autofluorescence imaging of NAD(P)H and FAD to assess changes in the metabolic activity of macrophages in response to tissue damage in live zebrafish. We show that these measurements detect metabolic changes in macrophages within interstitial tissue in response to sterile damage and microbial cues with temporal and spatial resolution. We also show that perturbations that modulate macrophage polarization and metabolism affect tissue repair.

## Results

### Autofluorescence imaging detects oxidized intracellular redox state in macrophages in vivo upon 2-deoxy-d-glucose treatment

To determine if a known glycolysis inhibitor alters macrophage metabolism in vivo, we imaged NAD(P)H and FAD in wounded zebrafish larvae in the absence and presence of 2-deoxy-d-glucose (2-DG). 2-DG is a glucose analog and acts as a competitive inhibitor of glycolysis at the step of phosphorylation of glucose by hexokinase (*Pelicano et al., 2006*). To isolate autofluorescence signals associated with macrophages from the whole tissue, we used mCherry and green fluorescent protein (GFP) transgenic reporter lines. GFP is suitable to image in conjunction with NAD(P)H, but it excludes the acquisition of FAD because they have overlapping spectra (*Datta et al., 2020*; *Qian et al., 2021*), while mCherry is compatible for simultaneous imaging with NAD(P)H and FAD (*Heaster et al., 2021*; *Hoffmann and Ponik, 2020*). The traditional serial acquisition of NAD(P)H and FAD was not suitable for imaging motile cells, such as macrophages, in live larvae. To accommodate cell movement during image acquisition in live larvae, we employed wavelength mixing that allows for simultaneous acquisition in three different channels (*Stringari et al., 2017*). We performed simple tail fin transection on transgenic larvae (*Tg(mpeg1:mCherry-CAAX)* that labels the plasma membrane of macrophages with mCherry), and performed autofluorescence imaging of NAD(P)H and FAD at the wound region (*Figure 1A*) at 3–6 hr post tail transection (hptt) in the absence or presence of 2-DG. As inhibiting glycolysis reduces NADH levels (*Georgakoudi and Quinn, 2012*; *Kolenc and Quinn, 2019*), we expected the optical redox ratio to decrease in macrophages of 2-DG-treated larvae compared to untreated control. Indeed, the optical redox ratio was significantly lower in macrophages in the 2-DG-treated larvae (*Figure 1B and C*). This change was driven by a decrease in NAD(P)H intensity in treated larvae, while FAD intensity remained similar to control levels (data not shown). Inhibition of glycolysis was associated with a significant reduction of the mean lifetime ($\tau_m$) of NAD(P)H in macrophages, with only a marginal reduction in FAD $\tau_m$ (*Figure 1D and E*). We also observed significant reduction for NAD(P)H and FAD $\tau_2$, and increase for NAD(P)H $\alpha_1$ (*Figure 1—figure supplement 1A-F*). These effects on NAD(P)H and FAD lifetime endpoints were similar to the effects observed with 2-DG treatment of activated T cells (*Walsh et al., 2021*). In sum, macrophages were more oxidized following treatment with a glycolysis inhibitor, and these findings support the utility of using autofluorescence imaging of metabolic coenzymes to detect changes in metabolic activity of macrophages in situ.

### Autofluorescence imaging detects metabolic changes in macrophages at the infected tail wound

To determine if autofluorescence imaging could distinguish different macrophage populations in a whole organism, we used a zebrafish *Listeria monocytogenes* (*Lm*)-infected tail wound model (*Miskolci et al., 2019*). This infection induces the recruitment of M1 macrophages as detected by a high level of TNFα expression (*Miskolci et al., 2019*). In contrast, most macrophages are devoid of

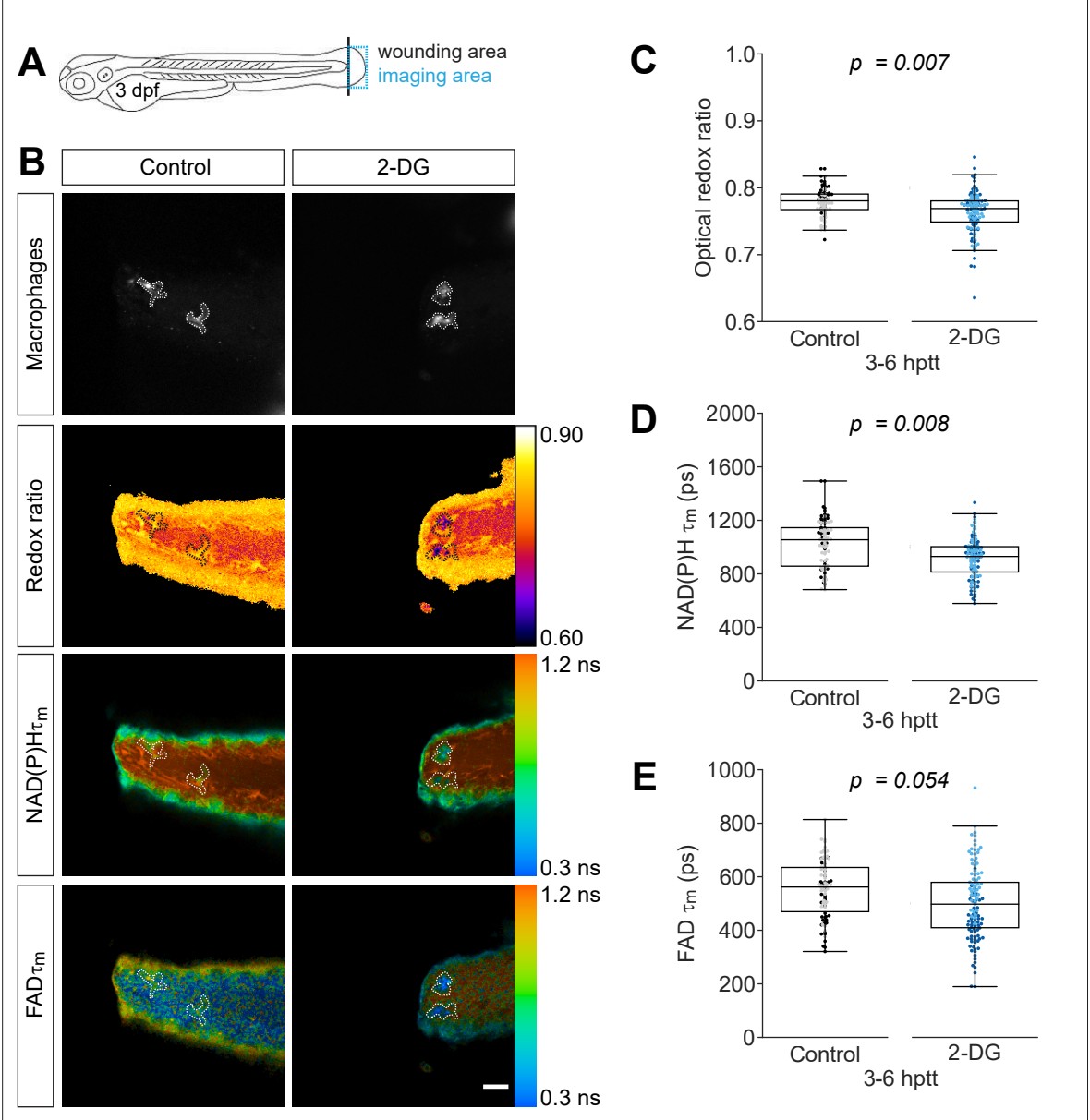

**Figure 1.** Inhibition of glycolysis reduces the optical redox ratio of macrophages following simple transection. Tail fin transection distal to the notochord was performed using transgenic zebrafish larvae (*Tg(mpeg1:mCherry-CAAX)*) that labels macrophages in the plasma membrane with mCherry at 3 days post fertilization, and autofluorescence imaging of nicotinamide adenine dinucleotide (phosphate) (NAD(P)H) and flavin adenine dinucleotide (FAD) was performed on live larvae at 3–6 hr post tail transection (hptt) that were either untreated (control) or treated with 5 mM 2-deoxy-d-glucose (2-DG) (glycolysis inhibitor) for 1 hr prior to imaging. (**A**) Schematic showing the area where wounding (black line) and imaging (blue box) were performed. (**B**) Representative images of mCherry (to show macrophages), optical redox ratio, and NAD(P)H and FAD mean lifetimes ($\tau_m$) are shown; macrophages in mCherry channel were outlined with dashed lines and the area was overlaid in the optical redox ratio and lifetime images to show corresponding location; scale bar = 50 µm. Quantitative analysis of (**C**) optical redox ratio, (**D**) NAD(P)H and (**E**) FAD mean lifetimes ($\tau_m$) from two biological repeats (control = 90 cells/9 larvae, 2-DG = 123 cells/9 larvae) is shown; quantitative analysis of associated individual lifetime endpoints ($\tau_1$, $\tau_2$, $\alpha_1$) and sample size for each repeat are included in *Figure 1—figure supplement 1*. The optical redox ratio and $\tau_m$ were log transformed prior to analysis. p values represent statistical analysis of the overall effects. Estimated means with 95% CI and overall effects with p values are included in *Figure 1—source data 1*.

The online version of this article includes the following source data and figure supplement(s) for figure 1:

**Source data 1.** Related to *Figure 1*.

**Figure supplement 1.** Individual nicotinamide adenine dinucleotide (phosphate) (NAD(P)H) and flavin adenine dinucleotide (FAD) fluorescence lifetime endpoints associated with *Figure 1*.

TNFα expression following a simple transection (*Miskolci et al., 2019*), and likely represent a differentially activated M2-like population (*Nguyen-Chi et al., 2015*).

Based on the differential activation profiles, we hypothesized that we would detect differences in the metabolic activity of macrophages at the simple and *Lm*-infected transection wounds. We performed tail fin transection in the absence or presence of *Lm* on double transgenic (*Tg(tnf:GFP) × Tg(mpeg1:mCherry-CAAX)*) larvae and performed autofluorescence imaging of NAD(P)H at the wound region on live larvae at 48 hr post wound (hpw). 48 hpw was chosen as a representative timepoint when the proportion of TNFα+ cells at these wounds was sufficiently different between the two wound models; most macrophages (~70%) are TNFα− in response to simple transection, whereas most macrophages (~80%) are TNFα+ at the infected wound, and these proportions do not significantly change at later timepoints (*Miskolci et al., 2019*). We performed autofluorescence imaging in conjunction with the TNFα reporter line (*tnf:GFP*) in order to monitor and group macrophages by

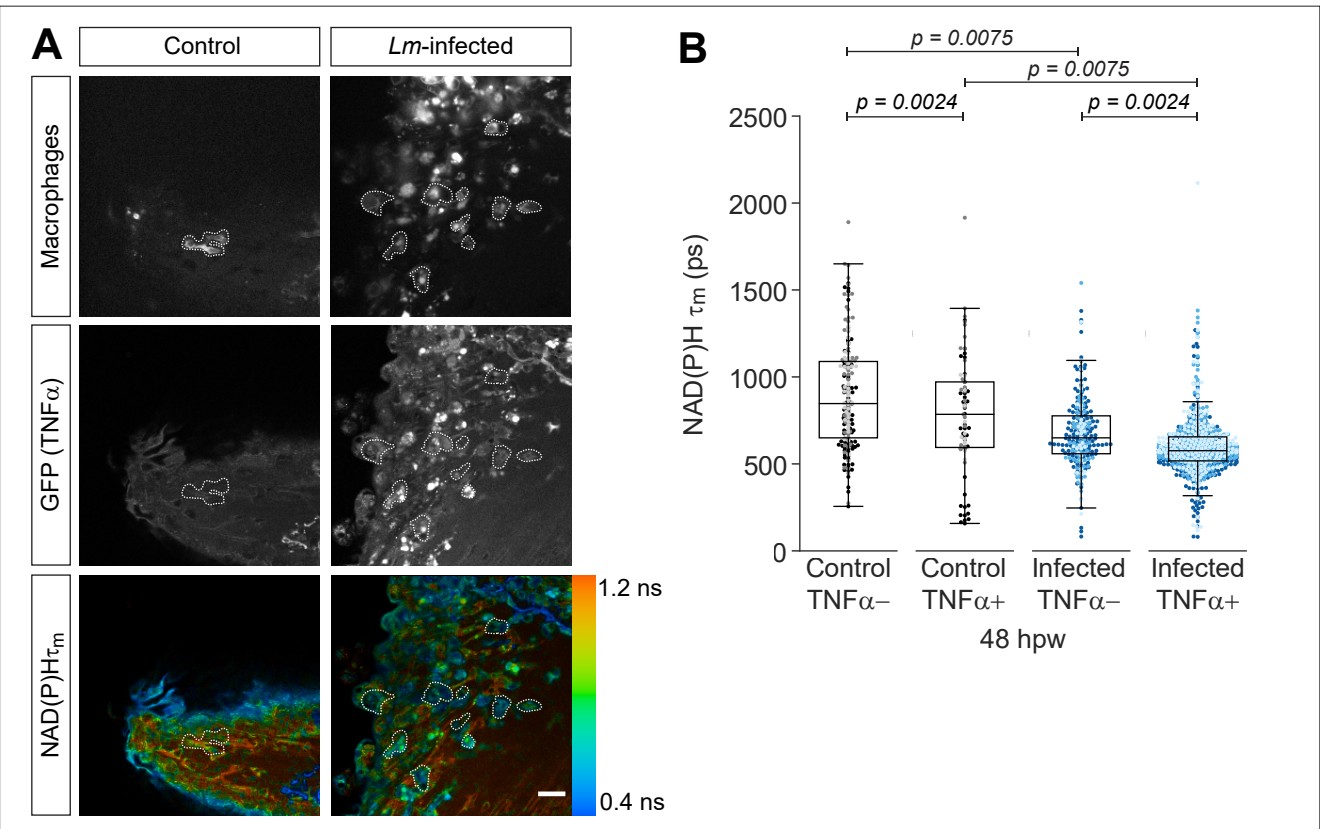

**Figure 2.** Mean lifetime of nicotinamide adenine dinucleotide (phosphate) (NAD(P)H) is reduced in TNFα+ macrophages at the infected tail wound. Tail fin transection distal to the notochord was performed using *N*-phenylthiourea-treated double transgenic zebrafish larvae (*Tg(tnf:GFP) x Tg(mpeg1:mCherry-CAAX)*), a TNFα reporter line in combination with a line that labels macrophages in the plasma membrane with mCherry) at 3 days post fertilization in the absence or presence of *Listeria monocytogenes (Lm)*. Autofluorescence imaging of NAD(P)H was performed on live larvae at 48 hr post wound (*Figure 1A*). (**A**) Representative images of mCherry expression to show macrophages, GFP to show TNFα expression, and NAD(P)H mean lifetime ($\tau_m$) are shown for control or *Lm*-infected tail wounds. Macrophages in the mCherry channel were outlined with dashed lines and the area was overlaid in GFP and lifetime images to show corresponding location; in the infected condition only a few macrophages are outlined as examples; scale bar = 50 µm. (**B**) Quantitative analysis of NAD(P)H mean lifetime ($\tau_m$) from three biological repeats (control TNFα− = 184 cells/16 larvae, control TNFα+ = 75 cells/16 larvae, infected TNFα− = 258 cells/16 larvae, infected TNFα+ = 789 cells/16 larvae) is shown; quantitative analysis of associated individual lifetime endpoints ($\tau_1, \tau_2, \alpha_1$) and sample size for each repeat are included in *Figure 2—figure supplement 1*. The $\tau_m$ was log transformed prior to analysis. Interaction between treatment and GFP expression was included to analyze whether either factor modified the effect of the other; no interaction was found. p values represent statistical analysis of the overall effects. Estimated means with 95% CI and overall effects with p values are included in *Figure 2—source data 1*.

The online version of this article includes the following source data and figure supplement(s) for figure 2:

**Source data 1.** Related to *Figure 2*.

**Figure supplement 1.** Individual nicotinamide adenine dinucleotide (phosphate) (NAD(P)H) fluorescence lifetime endpoints associated with *Figure 2*.

TNFα expression during image analysis. The TNFα reporter line relies on GFP expression to report transcriptional activity of *tnfα* (*Marjoram et al., 2015*), which precludes acquisition of FAD measurements. As a result, in this experiment we were not able to monitor changes in the intracellular optical redox ratio. Macrophages at the wound region were identified based on plasma membrane-localized mCherry expression as above. The infected tail wound recruited significantly more macrophages compared to the uninfected control (simple transection), and most macrophages at the infected tail wound expressed high levels of TNFα, while the majority lacked TNFα expression at the uninfected control wounds (*Figure 2A*, *Figure 2—figure supplement 1D*), consistent with our previous report (*Miskolci et al., 2019*). We detected a significant reduction in the mean lifetime ($\tau_m$) of NAD(P)H for TNFα+ macrophages relative to TNFα− macrophages in both the uninfected control and *Lm*-infected tail wounds (*Figure 2B*). While both types of wounds showed similar trends for TNFα+ and TNFα− macrophages, NAD(P)H $\tau_m$ was further reduced in macrophages from *Lm*-infected tail wounds relative to uninfected control when comparing either the TNFα− or TNFα+ groups (*Figure 2B*). Similar trends were observed for the individual lifetime components ($\tau_1, \tau_2$) of NAD(P)H as those observed for $\tau_m$ (*Figure 2—figure supplement 1A,B*), while we did not detect any significant changes in the fractional component of free NAD(P)H ($\alpha_1$) in any of the comparisons (*Figure 1—figure supplement 1C*).

Next, we repeated the same set of experiments but without the TNFα reporter, to acquire FAD measurements in order to monitor changes in the optical redox ratio. NAD(P)H $\tau_m$, $\tau_1$ and $\tau_2$ were significantly reduced in macrophages at the *Lm*-infected wound (*Figure 3D*, *Figure 3—figure supplement 1A,B*), consistent with the measurements above (*Figure 2B*, *Figure 2—figure supplement 1A,B*). We found that NAD(P)H $\alpha_1$ significantly increased in macrophages at the *Lm*-infected wound (*Figure 3—figure supplement 1C*). The presence of infection at the tail wound did not induce any significant changes in FAD lifetime endpoints (*Figure 3—figure supplement 1D-G*). Interestingly, the optical redox ratio was significantly reduced in macrophages at the highly inflammatory *Lm*-infected wound compared to the uninfected control, indicating that TNFα+ (M1-like) macrophage population is more oxidized compared to TNFα− (M2-like) in vivo (*Figure 3A and B*). The Optical Metabolic Imaging (OMI) index, a composite measure of the optical redox ratio, NAD(P)H $\tau_m$ and FAD $\tau_m$ (*Table 1*), was also lower in macrophages at the *Lm*-infected wound (*Figure 3C*). These findings were unexpected considering the observed increase of the optical redox ratio in the context of in vitro infection of bone marrow-derived macrophages (BMDM) with *Lm* (*Figure 3—figure supplement 1I,J*), indicating a more reduced intracellular redox state, consistent with previous publications of *Listeria* infection of macrophages in vitro (*Gillmaier et al., 2012*). We reasoned this result may be influenced by the presence of an intracellular pathogen in macrophages, and not solely due to a more proinflammatory macrophage phenotype. To test this, we next measured changes in the intracellular metabolism of macrophages in the context of thermal injury that also induces a TNFα+ macrophage population, but in the absence of infection.

## Autofluorescence imaging resolves changes in the metabolic activity of macrophages over the course of thermal tissue damage

To measure metabolic activity of macrophages during robust tissue damage, we used our zebrafish thermal injury tail wound model (*LeBert et al., 2018*). The burn wound elicits the recruitment of an M1-like macrophage population, as detected by TNFα expression (*Miskolci et al., 2019*). Unlike at the infected wound where TNFα expression in macrophages persists, TNFα+ macrophages peak at 24 hpw and resolve thereafter, as most macrophages at the wound are TNFα− by 72 hpw following thermal injury (*Miskolci et al., 2019*). We chose these two timepoints to compare the metabolic activity of macrophages in response to simple transection and thermal injury. Since macrophages are mostly TNFα− throughout the course of the wound response following a simple transection, we hypothesized that the metabolic activity of macrophages would be different at 24 hpw, but similar at 72 hpw, when comparing the two wound models.

We performed tail transection or generated a burn wound distal to the notochord on transgenic (*Tg(mpeg1:mCherry-CAAX)*) larvae, and performed autofluorescence imaging at the wound region on live larvae at 24 and 72 hpw. As expected, we observed significant differences in the metabolic activity of macrophages between the wounds at 24 hpw, but the cellular metabolism was similar at 72 hpw. Importantly, macrophages at the burn wound had a more oxidized redox state relative to macrophages at the simple transection at 24 hpw, indicated by the lower optical redox ratio and OMI index

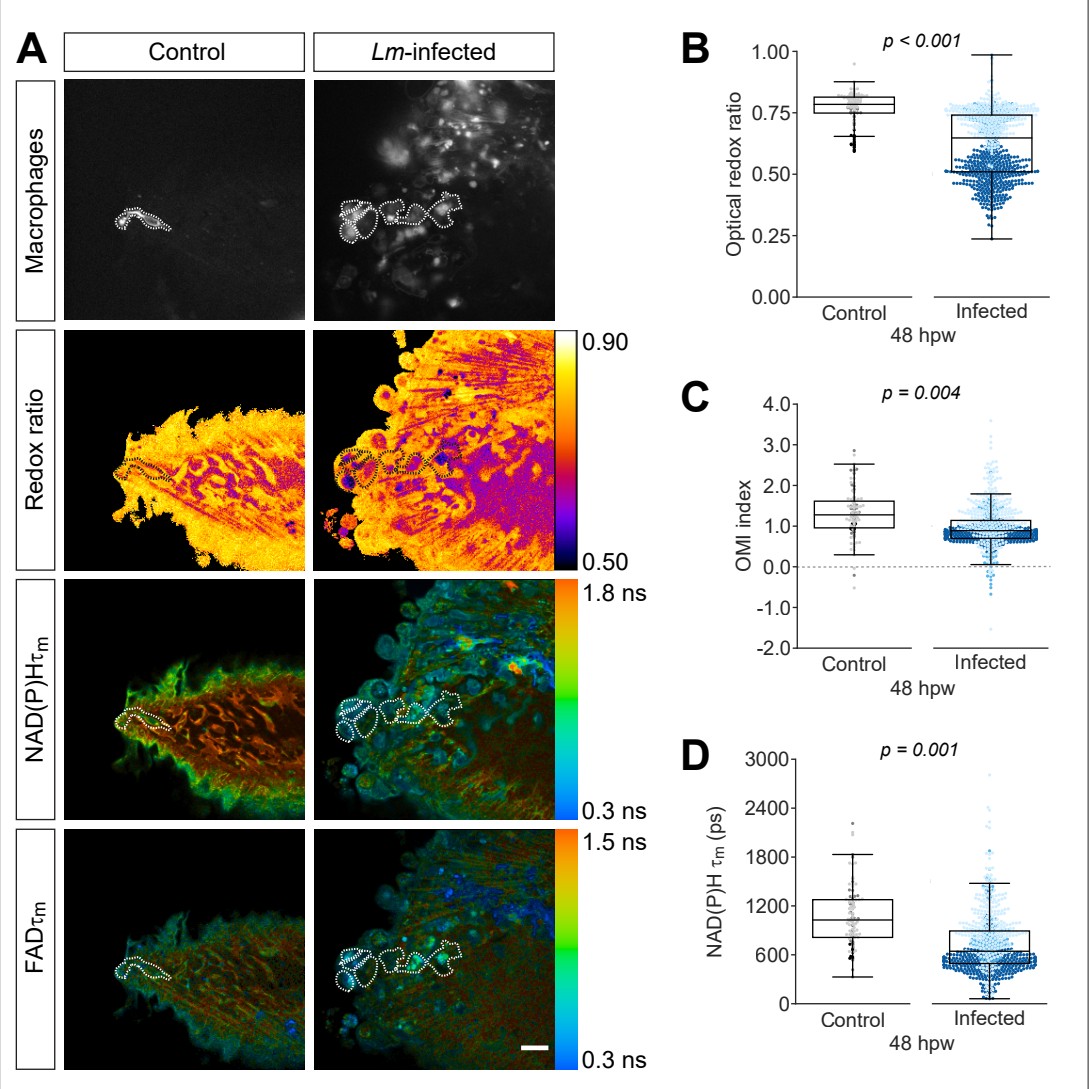

**Figure 3.** Optical redox ratio and mean lifetime of nicotinamide adenine dinucleotide (phosphate) (NAD(P)H) are reduced in macrophages at the infected tail wound. Tail fin transection distal to the notochord was performed using transgenic zebrafish larvae (*Tg(mpeg1:mCherry-CAAX)* that labels macrophages in the plasma membrane with mCherry) at 3 days post fertilization in the absence or presence of *Listeria monocytogenes* (*Lm*). Autofluorescence imaging of NAD(P)H and flavin adenine dinucleotide (FAD) was performed on live larvae at 48 hr post wound (*Figure 1A*). (**A**) Representative images of mCherry expression to show macrophages, optical redox ratio, and NAD(P)H and FAD mean lifetimes ($\tau_m$) are shown for control or infected tail wounds; macrophages were outlined with dashed lines and the area was overlaid in the optical redox ratio and lifetime images to show corresponding area; in the infected condition only a few macrophages are outlined as examples; scale bar = 50 µm. Quantitative analysis of (**B**) optical redox ratio, (**C**) Optical Metabolic Imaging index, and (**D**) NAD(P)H mean lifetime ($\tau_m$) from three biological repeats (control = 105 cells/16 larvae, infected = 761 cells/14 larvae) is shown; quantitative analysis of associated NAD(P)H and FAD mean ($\tau_m$) and individual lifetime endpoints ($\tau_1, \tau_2, \alpha_1$), and sample size for each repeat are included in *Figure 3—figure supplement 1*. p values represent statistical analysis of the overall effects. Estimated means with 95% CI and overall effects with p values are included in *Figure 3—source data 1*.

The online version of this article includes the following source data and figure supplement(s) for figure 3:

**Source data 1.** Related to *Figure 3*.

**Figure supplement 1.** Individual nicotinamide adenine dinucleotide (phosphate) (NAD(P)H) and flavin adenine dinucleotide (FAD) fluorescence lifetime endpoints associated with *Figure 3*.

(*Figure 4A–C*). In addition, NAD(P)H $\tau_m$ and $\tau_1$ were lower, while $\alpha_1$ was higher in macrophages at the burn compared to simple transection at 24 hpw (*Figure 4D*, *Figure 4—figure supplement 1A,C*); we did not detect any significant changes in FAD lifetime endpoints at 24 hpw (*Figure 4—figure supplement 1D-G*). Macrophages are mostly TNFα− at both wound types by 72 hpw (*Miskolci et al.,*

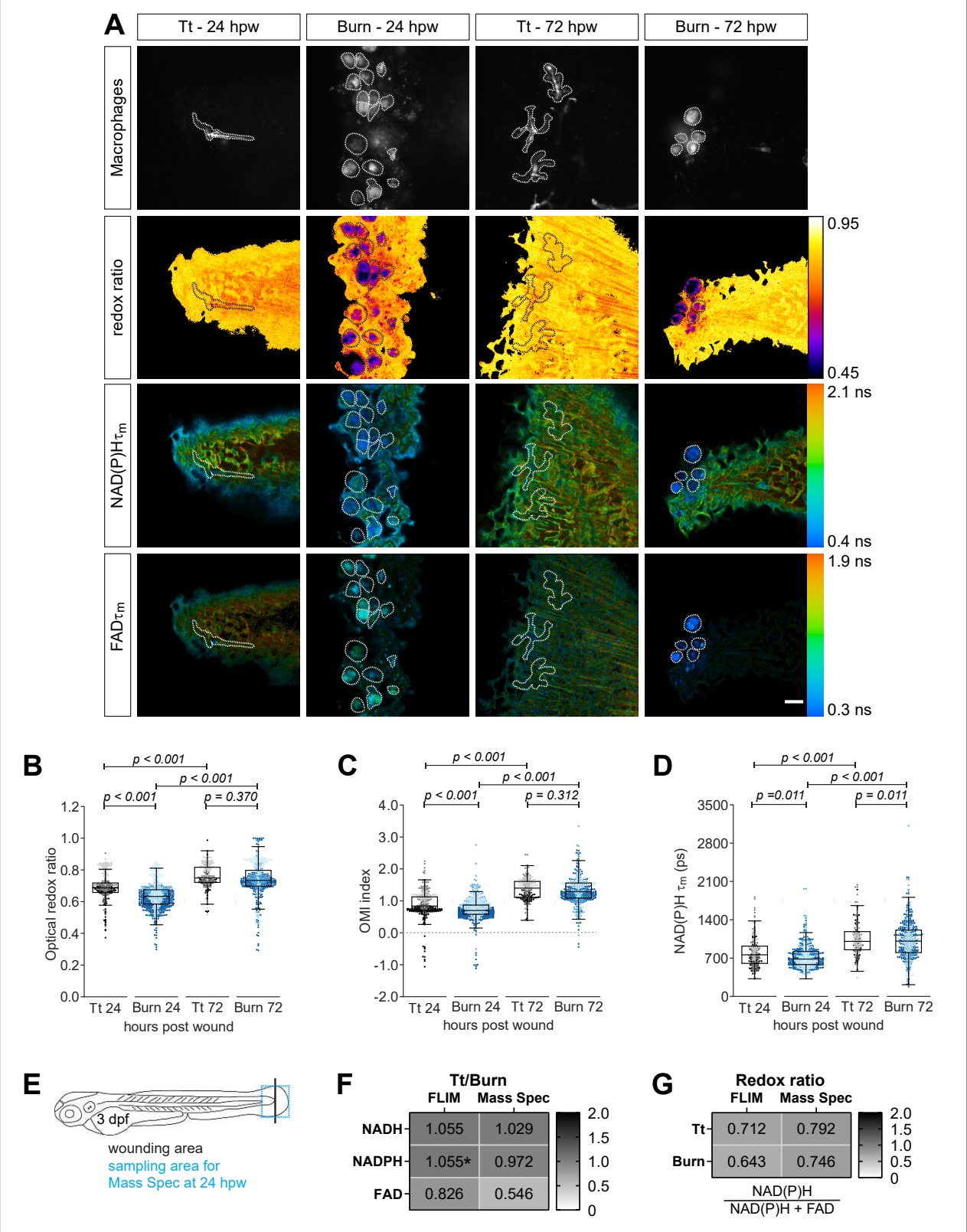

**Figure 4.** Autofluorescence imaging resolves temporal changes in the metabolic activity of macrophages at sterile tail wounds. Tail fin transection (Tt) or thermal injury (burn) distal to the notochord was performed using transgenic zebrafish larvae (*Tg(mpeg1:mCherry-CAAX)* that labels macrophages in the plasma membrane with mCherry) at 3 days post fertilization. Autofluorescence imaging of nicotinamide adenine dinucleotide (phosphate) (NAD(P)H) and flavin adenine dinucleotide (FAD) was performed on live larvae at 24 and 72 hr post wound (*Figure 1A*). (**A**) Representative images

*Figure 4 continued on next page*

*Figure 4 continued*

of mCherry expression to show macrophages, optical redox ratio, and NAD(P)H and FAD mean lifetimes ($\tau_m$) are shown for Tt or burn wounds; macrophages were outlined with dashed lines and the area was overlaid in the optical redox ratio and lifetime images to show corresponding location of macrophages; scale bar = 50 µm. Quantitative analysis of (**B**) optical redox ratio, (**C**) Optical Metabolic Imaging index, and (**D**) NAD(P)H mean lifetime ($\tau_m$) from three biological repeats (Tt-24 hr = 322 cells/16 larvae, burn-24 hr = 850 cells/14 larvae, Tt-72 hr = 213 cells/12 larvae, burn-72 hr = 578 cells/11 larvae) is shown; quantitative analysis of associated NAD(P)H and FAD mean ($\tau_m$) and individual lifetime endpoints ($\tau_1, \tau_2, \alpha_1$), and sample size for each repeat are included in *Figure 4—figure supplement 1*. Interaction between treatment and time was included to analyze whether either factor modified the effect of the other; strong interaction was detected for the optical redox ratio. p values represent statistical analysis of the overall effects. Estimated means with 95% CI and overall effects with p values are included in *Figure 4—source data 1*. (**E**) Tail fin tissue was collected distal to the caudal vein/artery loop (blue box) 24 hr following either tail transection or thermal injury distal to the notochord (black line) for mass spec analysis of small metabolites to compare the global trend of changes in redox metabolites with that measured by autofluorescence imaging; metabolomics data shown in (E) and (F) are from four biological repeats. (**F**) Metabolite abundance measured by either autofluorescence imaging or mass spec in transection sample was normalized by that in burn or (**G**) was used to calculate the redox ratio in transection (Tt) or burn samples. We included NADPH abundance in the redox ratio calculated using mass spec measurements. *NADPH and NADH intensities could not be collected separately by autofluorescence imaging as their fluorescence spectra overlap, thereby were measured collectively.

The online version of this article includes the following source data and figure supplement(s) for figure 4:

**Source data 1.** Related to *Figure 4*.

**Figure supplement 1.** Individual nicotinamide adenine dinucleotide (phosphate) NAD(P)H and flavin adenine dinucleotide (FAD) fluorescence lifetime endpoints associated with *Figure 4*.

*2019*), suggesting that the macrophage populations present at these wounds have similar activation states and are thereby likely to have similar metabolic activity. Accordingly, the optical redox ratio and OMI index were not different between macrophages of the simple transection and burn wound at 72 hpw (*Figure 4B and C*). The mean lifetime of NAD(P)H was significantly lower in macrophages at the burn wound relative to the simple transection at 72 hpw (*Figure 4D*), while it was similar for FAD (*Figure 4—figure supplement 1D*). Most of the individual lifetime endpoints ($\tau_1, \tau_2$ and $\alpha_1$) of NAD(P)H and FAD were also similar between the two wounds at 72 hpw (*Figure 4—figure supplement 1A-G*). Furthermore, we also detected temporal changes in the metabolic activity of macrophages during wound responses. The optical redox ratio and OMI index of macrophages increased over time at both wound types (*Figure 4B and C*), indicating a more reduced redox state. This would be expected as TNFα+ macrophages resolve at both wound types over time (*Miskolci et al., 2019*). In line with this, NAD(P)H $\tau_m, \tau_1$ and $\tau_2$ increased, while $\alpha_1$ decreased at both wound types over time (*Figure 4D*, *Figure 4—figure supplement 1A-C*). We also detected time-related changes in FAD endpoints; FAD $\tau_m$, and $\tau_1$ decreased over time at both wounds (*Figure 4—figure supplement 1D,E*). Collectively, we found that TNFα+ macrophage population was more oxidized, as indicated by a decrease in the redox ratio, and was associated with a decrease in NAD(P)H mean lifetime relative to a TNFα− macrophage population in context of both infected and sterile injury (*Table 2*).

To substantiate the metabolic changes observed by autofluorescence imaging in macrophages at the sterile tail wounds, we tested if we would see similar changes in tail fin tissue using targeted liquid

**Table 2.** Summary of changes in optical redox ratio, Optical Metabolic Imaging (OMI) index, and nicotinamide adenine dinucleotide (phosphate) (NAD(P)H) lifetime endpoints.
Changes in treated samples are shown relative to the control. Control, simple tail transection (uninfected); nd, not different.

| Control *versus* | 2-Deoxy-d-glucose | *Lm*-infected wound TNFα− | *Lm*-infected wound TNFα+ | *Lm*-infected wound | Burn wound 24 hr | Burn wound 72 hr |
|---|---|---|---|---|---|---|
| Optical redox ratio | ↓ | - | - | ↓ | ↓ | nd |
| OMI index | - | - | - | ↓ | ↓ | nd |
| NAD(P)H $\tau_m$ | ↓ | ↓ | ↓ | ↓ | ↓ | ↓ |
| NAD(P)H $\tau_1$ | nd | ↓ | ↓ | ↓ | ↓ | nd |
| NAD(P)H $\tau_2$ | ↓ | ↓ | ↓ | ↓ | nd | nd |
| NAD(P)H $\alpha_1$ | ↑ | nd | nd | ↑ | ↑ | ↑ |

chromatography-mass spectrometry (LS-MS)-based method to analyze the abundance of NADH, NAD(P)H, and FAD. We performed simple transection or burn wound distal to the notochord on unlabeled wild-type zebrafish larvae and collected the tail fin tissue distal to the caudal vein/artery loop (to remain close to the wound microenvironment) 24 hr following injury for targeted LC–MS metabolite analysis (*Figure 4E*). The technical limitation here is that we analyzed the abundance of these small metabolites in the whole tail fin tissue, not macrophages alone, because it is difficult to collect enough macrophages from such a small region to reach the detection limit of the mass spectrometer. We calculated the relative abundances of NAD(P)H and FAD in burn wound compared to transection, and found the trends measured by autofluorescence imaging and mass spectrometry to be consistent, in that the NAD(P)H levels are similar in both wound models, while FAD level is lower in transection (*Figure 4F*). Additionally, we compared the redox ratio measured in each wound model by both methods (*Figure 4G*). The two methods gave similar results, with both showing a trend toward a higher redox ratio in the transection model. These findings suggest that the redox state of macrophages and the whole tail fin tissue are similar after tissue damage.

## Metformin increases the optical redox ratio of wound macrophages

To further validate the use of autofluorescence imaging to characterize macrophage metabolism in vivo, we used metformin to modulate macrophage polarization. Metformin attenuates proinflammatory activation of macrophages in vitro and promotes an M2-like phenotype (*Schuiveling et al., 2018*). We previously reported that metformin also reduces the number of TNFα+ cells in zebrafish liver cancer models (*de Oliveira et al., 2019*). We found that metformin treatment reduced the proportion of TNFα+ macrophages at the burn wound, without decreasing the number of macrophages (*Figure 5A and B*, *Figure 5—figure supplement 1A*). We performed thermal injury of the tail fin on control or metformin-treated larvae and performed autofluorescence imaging on live larvae at 24 hr following injury. By decreasing TNFα+ cells, we expected to observe an increase in the optical redox ratio and NAD(P)H $\tau_m$. Indeed, we measured a significant increase in the optical redox ratio and OMI index of macrophages in metformin-treated larvae (*Figure 5C–E*). We also detected an increase in the mean lifetime of NAD(P)H, albeit not statistically significant (*Figure 5F*); NAD(P)H $\tau_1$ and $\tau_2$ also increased as expected, however, $\alpha_1$ did not change (*Figure 5—figure supplement 1B-D*). The modest changes in NAD(P)H lifetime endpoints are likely attributed to the small shift in macrophage polarization upon metformin treatment (*Figure 5B*).

## Macrophage metabolic switches are associated with changes in tissue repair

To determine if altering macrophage polarization and metabolic phenotype is associated with a change in wound healing, we treated thermal wounds with metformin or used STAT6-deficient zebrafish and characterized the effect on tissue regrowth. We first characterized the metabolic phenotype of macrophages in STAT6-depleted larvae, using a recently characterized *stat6* mutant (*Cronan et al., 2021*). STAT6 is a known regulator of M2-like macrophage polarization (*Murray, 2017*). In accordance with in vitro studies, we found that depletion of STAT6 led to an increase in the proportion of TNFα+ macrophages at the wound at 72 hr post burn (hpb) (*Figure 6A and B*), although there were fewer total macrophages at the wound (*Figure 6—figure supplement 1B*). Importantly, STAT6 depletion did not affect the total number of macrophages in whole larvae (*Figure 6—figure supplement 1A*). In accordance with the increase in TNFα+ macrophages at the wound, we detected a significant reduction in the optical redox ratio and NAD(P)H $\tau_m$ of macrophages in the *stat6* mutant larvae at 72 hpb (*Figure 6D and F*); the OMI index was also lower, albeit not statistically significant (*Figure 6E*). Although the proportion of TNFα+ macrophages was higher in the mutant larvae at 72 hpb, it was decreasing by 96 hpb (*Figure 6B*), suggesting that STAT6 depletion merely delays the switch from a TNFα+ macrophage population to TNFα− at the wound, but does not prevent it. To determine if this switch in the polarization of macrophages at the wound induces changes in wound healing, we quantified the impact of metformin and STAT6 depletion on regeneration after thermal injury. We found significantly improved wound healing in the presence of metformin, as indicated by the larger area of tissue regrowth (*Figure 6G and H*). By contrast, depletion of STAT6 resulted in more TNFα+ macrophages, reduced optical redox ratio, and impaired wound healing (*Figure 6I and J*). This is consistent with prior work suggesting that the presence of TNFα+ macrophages at the wound is associated with

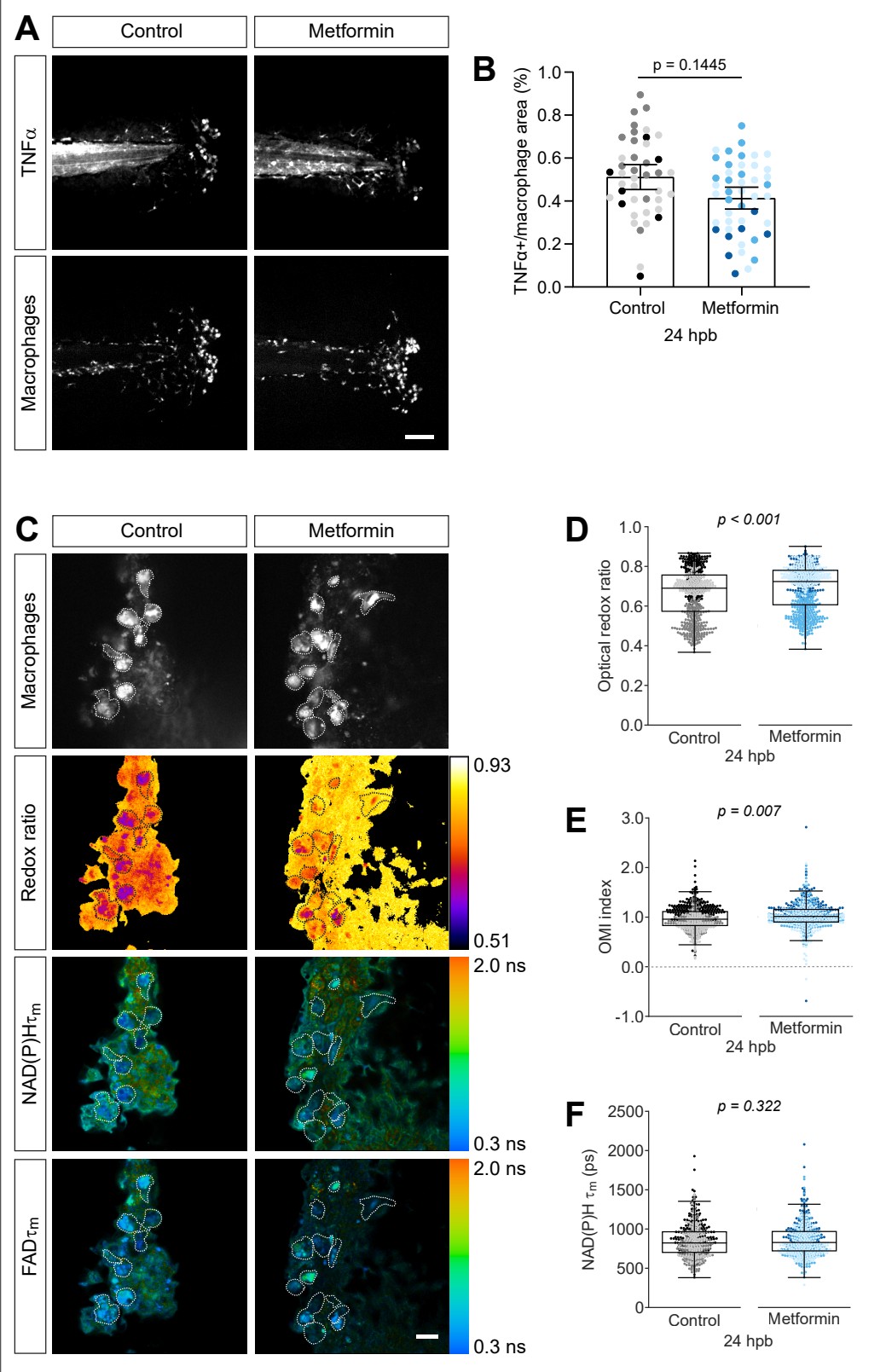

**Figure 5.** Metformin treatment increases optical redox ratio of macrophages. (**A, B**) Thermal injury (burn) distal to the notochord was performed using double transgenic zebrafish larvae (*Tg(tnf:GFP x mpeg1:mCherry-CAAX*) that labels macrophages in the plasma membrane with mCherry, and monitors TNFα expression using GFP-based TNFα reporter) at 3 days post fertilization (dpf), that were either untreated (control) or treated with

*Figure 5 continued on next page*

*Figure 5 continued*

1 mM metformin starting at 1 dpf; larvae were fixed at 24 hr post burn (hpb). (**A**) Representative images of sum projections of z-stacks acquired by spinning disk confocal microscopy are shown; scale bar = 100 μm. (**B**) TNFα expression in macrophages were quantified by area thresholding of GFP and mCherry intensities, and is displayed as proportional area per larva with bars showing arithmetic mean and 95% CI; results are from three biological repeats (control = 42, metformin = 46 larvae). (**C–F**) Thermal injury distal to the notochord was performed using control or metformin-treated transgenic zebrafish larvae (*Tg(mpeg1:mCherry-CAAX)*) at 3 dpf. Autofluorescence imaging of nicotinamide adenine dinucleotide (phosphate) (NAD(P)H) and flavin adenine dinucleotide (FAD) was performed on live larvae at 24 hpb. (**C**) Representative images of mCherry expression to show macrophages, optical redox ratio, and NAD(P)H and FAD mean lifetimes ($\tau_m$) are shown; scale bar = 50 μm. Quantitative analysis of (**D**) optical redox ratio, (**E**) Optical Metabolic Imaging index, and (**F**) NAD(P)H mean lifetime ($\tau_m$) from three biological repeats (control = 632 cells/13 larvae, metformin = 670 cells/14 larvae) is shown; quantitative analysis of associated NAD(P)H and FAD mean ($\tau_m$) and individual lifetime endpoints ($\tau_1, \tau_2, \alpha_1$), and sample size for each repeat are included in *Figure 5—figure supplement 1*. Log transformation was applied to optical redox ratio prior to analysis. p values represent statistical analysis of the overall effects. Estimated means with 95% CI and overall effects with p values are included in *Figure 5—source data 1*.

The online version of this article includes the following source data and figure supplement(s) for figure 5:

**Source data 1.** Related to *Figure 5*.

**Figure supplement 1.** Individual nicotinamide adenine dinucleotide (phosphate) (NAD(P)H) and flavin adenine dinucleotide (FAD) fluorescence lifetime endpoints associated with *Figure 5*.

impaired wound healing (*Krzyszczyk et al., 2018*; *Miskolci et al., 2019*). Taken together, these findings suggest that the metabolic activity of macrophages in the wound microenvironment correlates with the ability of damaged tissues to heal.

## Discussion

Understanding in vivo behavior has been limited by the lack of tools for the assessment of functional metabolic changes in live organisms. As a result, immunometabolism in vivo remains poorly characterized. Autofluorescense imaging of the endogenous fluorescence of metabolic coenzymes is an attractive approach because it allows for the quantitative analysis of metabolic changes on a single-cell level, while maintaining cells in their native microenvironment. Studies on the metabolic profiles of macrophages in vivo using autofluorescence imaging have been limited, with one study demonstrating that macrophages have distinguishable lifetime signatures from tumor cells in the tumor microenvironment (*Szulczewski et al., 2016*), and one study demonstrating changes in optical redox ratio, and mean lifetimes ($\tau_m$) of NAD(P)H and FAD between dermal and tumor macrophages in vivo in mice (*Heaster et al., 2021*).

Here, we took advantage of transparent zebrafish larvae to image macrophage metabolism in live animals with temporal and spatial resolution. We found that a proinflammatory (TNFα+) macrophage population was more oxidized relative to a TNFα− population, as indicated by a lower optical redox ratio and OMI index. The TNFα+ population was also associated with lower NAD(P)H $\tau_m, \tau_1$ and $\tau_2$, and these measurements were consistent across the proinflammatory wound models (*Table 2*). These results may reflect a reduction in glycolytic activity, as these changes are similar to what we found with 2-DG inhibition of glycolysis (*Table 2*).

In vitro infection of murine macrophages by *Lm* exhibited increased optical redox ratio relative to uninfected macrophages (*Figure 3—figure supplement 1I,J*), however, we found that infection of zebrafish larvae reduced the optical redox ratio in macrophages (*Figure 3B*). Surprisingly, we found that proinflammatory macrophages at sterile inflammatory sites were also associated with reduced optical redox ratios (*Figure 4B*), suggesting that in general, a proinflammatory macrophage population in vivo has a more oxidized redox state (*Table 2*). These findings suggest that macrophage metabolism in vitro and in vivo may differ and raises interesting questions for future investigation. Physiological functions in vitro do not always translate to in vivo settings; one example is the requirement of integrins for leukocyte migration on two-dimensional substrates in vitro, but it is dispensable for migration within three-dimensional interstitial tissues (*Lämmermann et al., 2008*). This metabolic variation most likely reflects the differences in the inherent nature of in vitro and in vivo microenvironments, such as interactions with other cells (*Van den Bossche and Saraber, 2018*). These findings

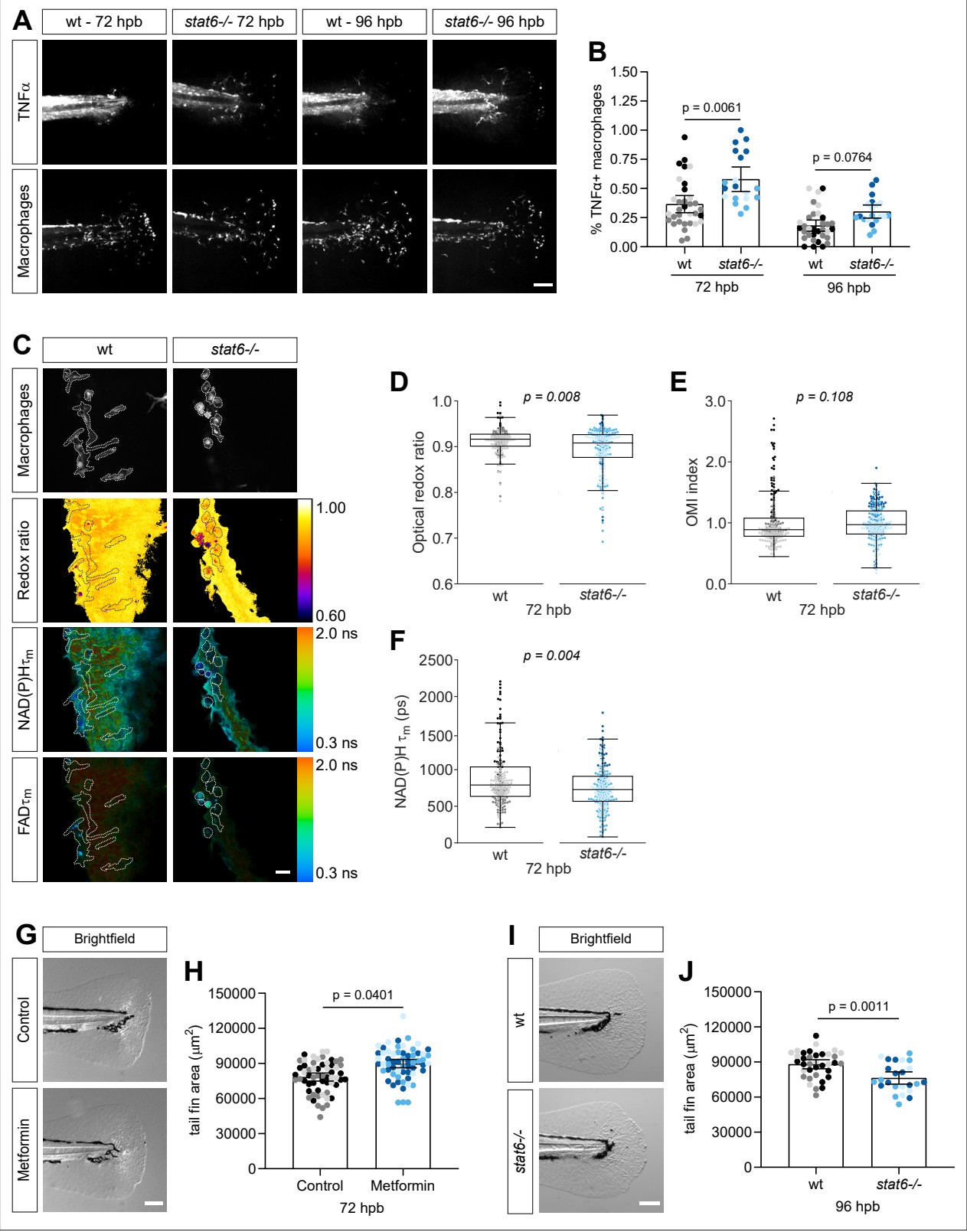

**Figure 6.** Macrophage metabolic switches are associated with changes in tissue repair. (**A–B**) Thermal injury distal to the notochord was performed using *N*-phenylthiourea-treated double transgenic wild-type (wt) or *stat6*-deficient zebrafish larvae (*Tg(tnf:GFP x mpeg1:mCherry-CAAX)*) at 3 days post fertilization (dpf); larvae were fixed at 72 and 96 hr post burn (hpb). (**A**) Representative images of sum projections of z-stacks acquired by spinning disk confocal microscopy are shown; scale bar = 100 μm. (**B**) TNFα expression in macrophages was quantified by scoring cells for GFP signal and is

*Figure 6 continued on next page*

*Figure 6 continued*

displayed as proportion of TNFα+ cells per larva with bars showing arithmetic mean and 95% CI; results are from three biological repeats (wt = 33, *stat6−/−* = 46 at 72 hpb, wt = 33, *stat6−/−* = 20 larvae at 96 hpb). Larvae were generated by a het incross and genotyped post imaging; only wt and *stat6−/−* larvae were analyzed. (**C–F**) Thermal injury distal to the notochord was performed using transgenic zebrafish larvae (*Tg(mpeg1:mCherry-CAAX)*) in wt- or *stat6*-deficient background at 3 dpf. Autofluorescence imaging of nicotinamide adenine dinucleotide (phosphate) (NAD(P)H) and flavin adenine dinucleotide (FAD) was performed on live larvae at 72 hpb. (**C**) Representative images of mCherry expression to show macrophages, optical redox ratio, and NAD(P)H and FAD mean lifetimes ($\tau_m$) are shown; macrophages were outlined with dashed lines and the area was overlaid in the optical redox ratio and lifetime images to show corresponding location of macrophages; scale bar = 50 μm. Quantitative analysis of (**D**) optical redox ratio, (**E**) Optical Metabolic Imaging index, and (**F**) NAD(P)H mean lifetime ($\tau_m$) from three biological repeats (wt = 217 cells/12 larvae, *stat6−/−* = 272 cells/12 larvae) is shown; quantitative analysis of associated NAD(P)H and FAD mean ($\tau_m$) and individual lifetime endpoints ($\tau_1, \tau_2, \alpha_1$), and sample size for each repeat are included in *Figure 6—figure supplement 1*. p values represent statistical analysis of the overall effects. Estimated means with 95% CI and overall effects with p values are included in *Figure 6—source data 1*. (**G**) Control or metformin-treated larvae were fixed at 72 hpb following thermal injury of the tail fin. Representative single-plane brightfield images are shown; scale bar = 100 μm. (**H**) Quantitative analysis of tail fin tissue regrowth area per larva, displayed with bars showing arithmetic mean and 95% CI. Results are from three biological repeats (control = 59, metformin = 60 larvae). (**I**) Wt- or *stat6*-deficient larvae were fixed at 96 hpb following thermal injury of the tail fin. Representative single-plane brightfield images are shown; scale bar = 100 μm. (**J**) Quantitative analysis of tail fin tissue regrowth area per larva, displayed with bars showing arithmetic mean and 95% CI. Results are from three biological repeats (wt = 36, *stat6−/−* = 24 larvae). Larvae were generated by a het incross and genotyped post imaging; statistical analysis was performed on wt, *stat6±* and *stat6−/−*; only wt and *stat6−/−* larvae are shown.

The online version of this article includes the following source data and figure supplement(s) for figure 6:

**Source data 1.** Related to *Figure 6*.

**Figure supplement 1.** Individual nicotinamide adenine dinucleotide (phosphate) NAD(P)H and flavin adenine dinucleotide (FAD) fluorescence lifetime endpoints associated with *Figure 6*.

underscore the importance of understanding macrophage metabolism directly in their native micro-environment and the development of new tools to probe these activities in situ.

We characterized macrophage populations in live zebrafish larvae using the TNFα reporter (*Marjoram et al., 2015*), a well-established marker of classically activated M1-like macrophages (*Murray, 2017*; *Nguyen-Chi et al., 2015*). The TNFα reporter has been used in several studies to identify M1-like macrophages in zebrafish (*de Oliveira et al., 2019*; *Miskolci et al., 2019*; *Nguyen-Chi et al., 2017*; *Nguyen-Chi et al., 2015*; *Roh-Johnson et al., 2017*). In light of the known in vitro metabolic profiles of macrophages, we expected a macrophage population with large numbers of TNFα+ cells to exhibit a more glycolytic state and thereby have higher redox ratio relative to a mostly TNFα− macrophage population. One caveat of intensity-based measurements is that other fluorophores, such as elastin and lipofuscin, could contribute to the intensity signals for the redox ratio and be a source of error (*Datta et al., 2020*). Another caveat is that NADH also exists in a phosphorylated form (NADPH) that is autofluorescent and has overlapping spectral properties with NADH *Blacker and Duchen, 2016*; hence for the sake of accuracy we use NAD(P)H to reflect their combined signal. The source and role of the observed oxidative metabolism in proinflammatory macrophages in the context of infection and sterile inflammation in vivo requires further analysis. Our results with metformin treatment, a drug known to inhibit the production of mitochondrial reactive oxygen species (mROS) at complex I of the electron transport chain in the mitochondria (*Schuiveling et al., 2018*), suggest that mROS contributes to the observed trends in the optical redox ratio and NAD(P)H lifetime profiles during sterile inflammation. It will be interesting to further explore the role of mROS in macrophage activation and function in vivo.

We found that a proinflammatory macrophage population is also associated with a decrease in mean lifetime of NAD(P)H (*Table 2*). Similarly, inhibition of metabolic reactions in the cell (e.g. rotenone for oxidative phosphorylation, 2-DG for glycolysis) decreases NAD(P)H $\tau_m$ in vitro and in vivo (*Walsh et al., 2021*; *Yaseen et al., 2017*). Weights are given to the free ($\alpha_1$) and bound ($\alpha_2$) components, so that pools of NAD(P)H in the free or bound state can be quantified. The lifetimes themselves ($\tau_1, \tau_2$) are affected by preferred protein-binding activities in the cell ( >300 proteins bind to NAD(P)H in the cell *Berman et al., 2000*), and microenvironmental factors (e.g. pH, viscosity, and temperature). Therefore, the NAD(P)H lifetime is a unique biophysical measurement that is influenced by the amount of NAD(P)H in the free and bound pools, preferred protein-binding activities in the cell, and microenvironmental factors. In the current study, we report the mean lifetime of NAD(P)H ($\tau_m$) as the weighted average of the short and long lifetime components ($\tau_m = \alpha_1\tau_1 + \alpha_2\tau_2$). Therefore, a decrease in $\tau_m$ is due to an increase in the pool of free NAD(P)H, a decrease in $\tau_1$, and/or a decrease in $\tau_2$ (*Table 1*).

Interestingly, we found that TNFα expression in macrophages at the control and infected tail wounds was associated with a graded effect on NAD(P)H lifetime endpoints that was on continuum (*Figure 2B*, *Figure 2—figure supplement 1A-C*). TNFα– macrophages from uninfected wound (control) are on one end of the spectrum, while TNFα+ cells from the infected wound are on the opposite end. As we move on this spectrum, we see a graded change in the mean and individual lifetime endpoints in the same direction from one end to the other end, reminiscent of the concept that macrophage activation in vivo occurs in a continuum of activation states as opposed to a more strict M1 or M2 classification (*Murray, 2017*). These results suggest that autofluorescence imaging is sensitive to detect variations in macrophage populations across different levels of activation. The single cell-based approach is one key advantage of autofluorescence imaging as it allows for analyzing metabolic heterogeneity in a cell population. In future studies we will apply distribution density models as in *Heaster et al., 2019*; *Walsh et al., 2021* to monitor heterogeneity in macrophages during wound responses.

Our findings show that autofluorescence imaging is also capable of resolving time-related changes in macrophage metabolism. We observed that the optical redox ratio and OMI index increased in macrophages over time, indicating that the intracellular redox state becomes more reduced over time, both at the simple transection and the burn wound (*Figure 4B and C*). This was expected based on our previous report that the early macrophage population at the burn wound is mostly TNFα+, however, over time the macrophage population becomes mostly TNFα–, similar to the simple transection, and this switch in activation phenotype coincided with a recovery in wound healing (*Miskolci et al., 2019*). The observed increase in the optical redox ratio over time is interesting. Macrophages polarize toward a prohealing M2-like state during wound healing, thus based on existing literature macrophages would be expected to rely on oxidative metabolism (*Caputa et al., 2019*; *O'Neill et al., 2016*) and thereby display a reduction in the optical redox ratio. However, it has been demonstrated that M2-like macrophages are more motile compared to M1-like cells (*Hind et al., 2016*) and glycolytic reprogramming has been shown to be important for macrophage migration (*Semba et al., 2016*). These reports suggest that our observed increase in the optical redox ratio at the wound, reflecting an increase in glycolytic activity, may be supporting the more motile nature of prohealing M2-like cells, however, this requires further investigation. Nevertheless, our data suggests that a reduced intracellular redox state in macrophages supports wound healing; this is supported by the results of the metformin treatment showing that an increase in the optical redox ratio (*Figure 5D*) is associated with improved wound healing (*Figure 6H*) and the converse by Stat6 depletion (*Figure 6D and J*). Our results are in line with recent zebrafish studies that found that tail transection leads to a shift to glucose metabolism and wound healing is blocked upon 2-DG treatment (*Sinclair et al., 2021*), and progenerative macrophages display a glycolytic phenotype in context of muscle injury as indicated by increased levels of intracellular NADH measured by two-photon autofluorescence imaging and bioluminescence-based assay (*Ratnayake et al., 2021*).

Finally, our findings demonstrate a correlation between macrophage M1 phenotype and impaired wound healing. Treatment with metformin reduced the TNFα expression in macrophages and improved wound healing. By contrast, depletion of STAT6 increased TNFα expression and impaired wound healing (*Figure 6*). It will be interesting to determine if macrophage intrinsic STAT6 is sufficient to regulate its metabolic activity and the fate of tissue repair.

With the emergence of immunometabolism it is now recognized that metabolic reprogramming underlies macrophage activation and function. Differential activation of macrophages plays a central role in host health and disease progression, underscoring the importance of studying macrophage metabolism in vivo. We have shown that fluorescence intensity and lifetime imaging of NAD(P)H and FAD can resolve metabolic changes in macrophages with distinct activation states in situ in a live organism, suggesting that this approach can be a valuable label-free imaging-based tool to study the metabolic regulation of immune cell function in vivo.

# Materials and methods

**Key resources table**

| Reagent type (species) or resource | Designation | Source or reference | Identifiers | Additional information |
|---|---|---|---|---|
| Strain, strain background (10,403 S) | *Listeria monocytogenes* (strain 10,403 S) | PMID:26468080 | | Was be obtained from JD Sauer Lab, University of Wisconsin - Madison |

| Reagent type (species) or resource | Designation | Source or reference | Identifiers | Additional information |
|---|---|---|---|---|
| Strain, strain background (*Danio rerio*) | WT (AB) | ZIRC | ZL1 | https://zebrafish.org/home/guide.php |
| Strain, strain background (*D. rerio*) | *Tg(tnf:GFP)* (AB) | PMID:25730872 | | Was obtained from Michel Bagnat Lab, Duke University |
| Strain, strain background (*D. rerio*) | *Tg(mpeg1:histone2b-GFP)* (AB) | PMID:31259685 | | Can be obtained from Anna Huttenlocher Lab, University of Wisconsin - Madison |
| Strain, strain background (*D. rerio*) | *Tg(mpeg1:mCherry-CAAX)* (albino) | PMID:26887656 | | Was obtained from Leonard Zon Lab, Boston Children's Hospital, Dana Farber Cancer Institute |
| Strain, strain background (*D. rerio*) | *stat6* mutant (AB) | PMID:33761328 | | Was obtained from David Tobin Lab, Duke University |
| Chemical compound, drug | Metformin | Enzo Life Sciences, cat no ALX-270–432 G005 | https://www.enzolifesciences.com/ALX-270-432/metformin/ | 1 mM in E3, bathing, start treatment at 1 day post fertilization, refresh daily |
| Chemical compound, drug | 2-Deoxy-d-glucose | Sigma, cat no D8375 | https://www.sigmaaldrich.com/US/en/product/sigma/d8375 | 5 mM in E3, bathing, 1 hr pretreatment right before imaging |
| Software, algorithm | GraphPad Prism | | RRID:SCR_002798 | https://www.graphpad.com/scientific-software/prism/ |
| Software, algorithm | SAS | | RRID:SCR_008567 | https://www.sas.com/en_us/home.html |
| Software, algorithm | Fiji, ImageJ | *Schindelin et al., 2012* | RRID:SCR_002285 | https://fiji.sc/ |
| Software, algorithm | Cell profiler | | RRID:SCR_007358 | https://cellprofiler.org/ |
| Software, algorithm | Matplotlib | | RRID:SCR_008624 | https://matplotlib.org/ |
| Software, algorithm | R project for statistical computing | *R Core Team, 2019* | RRID:SCR_001905 | https://www.r-project.org/ |
| Software, algorithm | MATLAB R2019b | | | https://www.mathworks.com/ |
| Software, algorithm | SPCImage 7.4 | | | https://www.becker-hickl.com/products/spcimage/ |
| Other | Line-powered thermal cautery instrument | Stoelting | 59,005 | https://stoeltingco.com/Neuroscience/Thermal-Cautery-Instruments~9879 |
| Other | Type E tip for cautery instrument | Stoelting | 59,010 | https://stoeltingco.com/Neuroscience/Thermal-Cautery-Instruments~9879 |

## Zebrafish husbandry

All protocols using zebrafish and mice in this study have been approved by the University of Wisconsin-Madison Research Animals Resource Center (protocols M005405-A02/zebrafish, M005916/mouse). Adult zebrafish were maintained on a 14 hr:10 hr light/dark schedule. Upon fertilization, embryos were transferred into E3 medium and maintained at 28.5°C. To prevent pigment formation, larvae were maintained in E3 medium containing 0.2 mM *N*-phenylthiourea (PTU) (Sigma-Aldrich, St. Louis, MO) starting at 1 day post fertilization (dpf). Adult wild-type zebrafish, transgenic lines *Tg(tnf:GFP)* (*Marjoram et al., 2015*), and *Tg(mpeg1:histone2b-GFP)* (*Miskolci et al., 2019*) in wild-type or *stat6*-deficient zebrafish (*Cronan et al., 2021*) in AB background, and *Tg(mpeg1:mCherry-CAAX)* (*Bojarczuk et al., 2016*) in wild-type or *stat6*-deficient zebrafish in albino background were utilized in this study. To genotype *stat6*-deficient zebrafish, genomic DNA was used in a GoTaq Green PCR reaction (cat no. M7123; Promega, Madison, WI) with forward primer 5'-TATGCAGTTCCCTCCCTTCG-3' and reverse primer 5'-AGCTGATGAAGT-GTTTGGCG-3' (*Cronan et al., 2021*); PCR products were resolved by a 3% MetaPhor agarose gel (Lonza Rockland, Rockland, ME).

## Bacterial culture and preparation

Unlabeled or mCherry-expressing *Lm* strain 10,403 was used in this study (*Vincent et al., 2016*). *Lm* were grown in brain–heart infusion (BHI) medium (Becton, Dickinson and Company, Sparks, MD). A streak plate from frozen stock was prepared and grown overnight at 37°C; the plate was stored at 4°C. The day before infection, a fresh colony was picked from the streak plate and grown statically in 1 mL BHI overnight at 30°C to reach stationary phase and to flagellate bacteria. The next day bacteria were prepared to infect either primary macrophages or zebrafish larvae. To prepare for infection of primary cells, the 1 mL suspension was diluted with 1 mL sterile PBS, OD was determined to calculate the number of bacteria to infect cells at multiplicity of infection (MOI) of 2 (1 cell:2 bacteria) (OD 1 = $7.5 \times 10^8$ bacteria). To prepare for zebrafish tail wound infection, bacteria were subcultured for ~1.5–2.5 hr in fresh BHI (1:4 culture:BHI; 5 mL total) to achieve growth to midlogarithmic phase (OD 600 ≈ 0.6–0.8). From this subcultured bacterial suspension, 1 mL aliquot was collected, spun down at high speed for 30 s at room temperature, washed three times in sterile PBS, and resuspended in 100 µL of sterile PBS.

## *Lm* infection of mouse bone marrow-derived macrophages

About 6- to 8-week-old C57BL/six female mice were obtained from NCI/Charles River NCI facility and BMDM were made as previously described (*Sauer et al., 2011*). Briefly, macrophages were cultured from bone marrow in the presence of M-CSF derived from transfected 3T3 cell supernatant for 6 days, with an additional supplement of M-CSF medium 3 days postharvest. Cells were frozen down for storage. The day before infection, frozen cells were thawed and plated in 35 mm glass bottom dishes (MatTek, Ashland, MA) at $1.6 \times 10^6$ in 2.4 mL BMDM medium (Roswell Park Memorial Institute (RPMI) media containing 10% fetal bovine serum, 10% CSF, 1% sodium pyruvate, 1% glutamate, and 0.1% β-mercaptoethanol) and allowed to recover overnight at 37°C, 5% $CO_2$. The following day 1.6 mL BMDM medium with or without *Lm* at MOI 2 was added to cells and incubated at 37°C and 5% $CO_2$. After 30 min, cells were rinsed once with BMDM medium and replaced with 2.4 mL medium containing 0.25 mg/mL gentamicin (Lonza, Walkersville, MD). Cells were maintained at 37°C and 5% $CO_2$ and imaged live at 5–6 hr post infection.

## Fixation

Larvae were fixed in 1.5% formaldehyde (Polysciences, Warrington, PA) in 0.1 M pipes (Sigma-Aldrich), 1.0 mM $MgSO_4$ (Sigma-Aldrich), and 2 mM ethylene glycol tetraacetic acid (EGTA) (Sigma-Aldrich) overnight at 4°C, with gentle rocking. Next day samples were rinsed with PBS at least once and stored in PBS at 4°C until imaging.

## Counting total number of macrophages in whole larvae

To count macrophages, the nucleus was labeled using transgenic line *Tg(mpeg1:h2b-GFP)*. Larvae at 3 dpf (not treated with PTU) were fixed overnight at 4°C using 1.5% fixative. The z-stack images of whole larvae were acquired at room temperature using Zeiss Zoomscope (EMS3/SyCoP3; Zeiss, Oberkochen, Germany; Plan-NeoFluar Z objective; 25× magnification, 1.7 µm resolution, 9.2 mm field of view, 60 µm depth of field; 10 µm steps in z direction) and Zen software (Zeiss). Sum projections were generated and macrophages were counted manually using open source image processing package, Fiji (*Schindelin et al., 2012*).

## Zebrafish tail fin wounding

Simple tail fin transection, infected tail fin transection, and thermal injury of the tail fin were performed on 3 dpf larvae as described previously (*Miskolci et al., 2019*). In preparation for wounding, larvae were anesthetized in E3 medium containing 0.16 mg/mL tricaine (ethyl 3-aminobenzoate; Sigma-Aldrich). Wounding was performed in plastic tissue culture dishes that were coated with milk to prevent larvae sticking to the plastic. Simple tail transection of the caudal fin was performed using surgical blade (Feather, no. 10) at the boundary of and without injuring the notochord; following transection, larvae were rinsed with E3 medium to wash away tricaine, placed in fresh milk-coated dishes with fresh E3 medium, and maintained at 28.5°C until live imaging. For infected tail transection, larvae were placed in 5 mL E3 medium containing tricaine in 60 mm milk-treated dish; 100 µL unlabeled bacterial suspension in PBS, or 100 µL PBS for control uninfected wounding, was added

to the E3 medium and swirled gently to achieve even distribution of bacteria; tail fin transection of larvae in control or infected E3 medium was performed as described above; larvae were immediately transferred to a horizontal orbital shaker and shaken for 30 min at 70–80 rpm; control and infected larvae were then rinsed five times with 5 mL E3 medium without tricaine to wash away bacteria and maintained at 28.5°C until live imaging; larvae were not treated with antibiotics at any point during the experiment. To perform thermal injury, fine tip (type E) of a line-powered thermal cautery instrument (Stoelting, Wood Dale, IL) was placed into the E3 medium, held to the posterior tip of the caudal fin, and turned on for 1–2 s until tail fin tissue curled up without injuring the notochord; following injury, larvae were rinsed with E3 medium to wash away tricaine, placed in fresh milk-coated dishes with fresh E3 medium, and maintained at 28.5°C until live imaging or fixation.

## Drug treatments

2-DG (Sigma-Aldrich, St. Louis, MO) treatment was empirically optimized by testing different doses and length of pretreatment. 2-DG was freshly prepared for each experiment, dissolved at 100 mM in E3 medium without methylene blue (E3−); treatment was performed by bathing zebrafish larvae in E3− containing 5 mM 2-DG for 1 hr before imaging; larvae were kept in the presence of the inhibitor during live imaging. Metformin (Enzo, Farmingdale, NY) was freshly dissolved at 50 mM in E3−; treatment was performed by bathing dechorionated larvae in E3− containing 1 mM metformin starting at 1 dpf; metformin was refreshed daily (prepared fresh stock and dilution daily) until completion of live imaging or fixation.

## Embedding zebrafish larvae for live imaging

Larvae were embedded in 1 mL 1% low gelling agarose (Sigma-Aldrich) prepared in E3− in Ibidi μ-slide 2-well glass bottom chamber (Ibidi, Fitchburg, WI) and topped off with 1 mL E3−. Agarose and top-off solution were supplemented with 0.16 mg/mL tricaine to keep larvae anesthetized during live imaging. In drug treatment experiments, agarose and top-off solution were supplement with drugs at working concentrations.

## Wound healing assay

Wound healing assay was performed as previously (*Miskolci et al., 2019*). Larvae were wounded at 3 dpf as above and maintained at 28.5°C until fixation at indicated times. Fixed samples were placed in plastic milk-coated tissue culture dish in 0.1% Tween-20-PBS solution and tails were cut past the tip of the yolk sac so that tail fin tissue would lay flat. Single-plane brightfield images were acquired using Zeiss Zoomscope (EMS3/SyCoP3; Zeiss, Oberkochen, Germany; Plan-NeoFluar Z objective; 112× magnification, 0.7 μm resolution, 2.1 mm field of view, 9 μm depth of field) and Zen software (Zeiss). Tissue regrowth area as a measure of wound healing was quantified using Fiji by outlining the tail fin tissue area distal to the notochord using the polygon tool.

## TNFα expression in macrophages

Analysis of TNFα expression in macrophages at zebrafish tail wound was performed as previously (*Miskolci et al., 2019*) using TNFα reporter line (*Marjoram et al., 2015*). Double transgenic line *Tg(tnf:GFP x mpeg1:mCherry-CAAX)* was wounded at 3 dpf as above and maintained at 28.5°C until fixation at indicated times. Fixed samples were placed in Ibidi μ-slide 2-well glass bottom chamber in 0.1% Tween-20-PBS solution and z-stacks at 3 μm steps and 512 × 512 resolution were acquired at room temperature using a spinning disk confocal microscope (CSU-X, Yokogawa, Sugar Land, TX) with a confocal scanhead on a Zeiss Observer Z.1 inverted microscope, EC Plan-Neofluar NA 0.3/10× objective, a Photometrics Evolve EMCCD camera and Zen software (Zeiss). TNFα expression was quantified by scoring macrophages at the wound for GFP signal (TNFα− in the absence of GFP signal or TNFα+ when any GFP signal was detected within a cell), or by area thresholding in Fiji as previously described (*Miskolci et al., 2019*), and expressed as proportion per larva.

## Fluorescence lifetime imaging of NAD(P)H and FAD

All samples were imaged live using a two-photon fluorescence microscope (Ultima, Bruker) coupled to an inverted microscope body (TiE, Nikon), adapted for fluorescence lifetime acquisition with time correlated single photon counting electronics (SPC-150, Becker & Hickl, Berlin, Germany). A 40× (NA = 1.15) water immersion objective was used. An Insight DS+ (Spectra Physics) femtosecond source with dual emission provided light at 750 nm (average power: 1.4 mW) for NAD(P)H excitation and 1040 nm (average power: 2.1 mW; for *stat6* mutant experiment the average power was 5 mW to compensate for an excitation intensity drop caused by an underfilled objective resulting from adjustments to achieve a more uniform mixed wavelength excitation over a larger field of view) for mCherry excitation. FAD excitation at 895 nm was achieved through wavelength mixing. Wavelength mixing was achieved by spatially and temporally overlapping two synchronized pulse trains at 750 nm and 1040 nm (*Stringari et al., 2017*). Bandpass filters were used to isolate light, with 466/40 nm used for NAD(P)H and 540/24 nm for FAD, and 650/45 for mCherry which were then detected by GaAsP photomultiplier tubes (H7422, Hamamatsu). Fluorescence lifetime decays of NAD(P)H, FAD, and mCherry were acquired simultaneously with 256 time bins across 256 × 256 pixel images within Prairie View (Bruker Fluorescence Microscopy) with a pixel dwell time of 4.6 μs and an integration time of 60 s (image acquisition time) at an optical zoom of 2.00. No change in the photon count rate was observed, ensuring that photobleaching did not occur. Images were acquired in a single plane. Following each acquisition, we moved to a different plane to avoid measuring the same cell twice; each data point in the graphs represents a different macrophage. The second harmonic generation obtained from urea crystals excited at 890 nm was used as the instrument response function and the full width at half maximum was measured to be 260 ps. BMDM were imaged live in MatTek dishes while maintained at 37°C and 5% $CO_2$ using a stage top incubator system (Tokai Hit, Bala Cynwyd, PA). Zebrafish larvae were embedded in 1% low-gelling agarose and imaged live at room temperature; double transgenic larvae *Tg(tnf:GFP × mpeg1:mCherry-CAAX)* were PTU treated, all other imaging was done in albino background.

## Fluorescence lifetime data analysis

Fluorescence lifetime components were computed in SPCImage v7.4 (Becker and Hickl). For each image, a threshold was selected to exclude background. The fluorescence lifetime components were then computed for each pixel by deconvolving the measured instrument response function and fitting the resulting exponential decay to a two-component model, $I(t) = \alpha_1 e^{t/\tau_1} + \alpha_2 e^{t/\tau_2} + C$, where I(t) is the fluorescence intensity at time t after the laser excitation pulse, $\alpha_1$ and $\alpha_2$ are the fractional contributions of the short and long lifetime components, respectively (i.e. $\alpha_1 + \alpha_2 = 1$), $\tau_1$ and $\tau_2$ are the fluorescence lifetimes of the short and long lifetime components, respectively, and C accounts for background light. A two-component decay was used to represent the lifetimes of the free and bound configurations of NAD(P)H and FAD (*Lakowicz et al., 1992*; *Nakashima et al., 1980*). Images were analyzed at the single cell level. For the in vitro macrophages, cell cytoplasm masks were obtained using a custom CellProfiler pipeline (v.3.1.8) (*McQuin et al., 2018*). Briefly, the user manually outlined the nucleus of the cells and those masks were then propagated outward to find cell areas. Cytoplasm masks were then determined by subtracting the nucleus masks from the total cell area masks. Bacteria masks were created in Fiji by thresholding the mCherry intensity images into bacteria and background. The resulting bacteria masks were then subtracted from the corresponding field of view's masks to exclude bacterial metabolic data. The diffuse cytoplasmic fluorescence in the mCherry images is likely due to FAD autofluorescence (*Szulczewski et al., 2016*). Images of the optical redox ratio (intensity of NAD(P)H divided by the sum of the intensity of NAD(P)H and the intensity of FAD) and the mean fluorescence lifetime ($\tau_m = \alpha_1 \tau_1 + \alpha_2 \tau_2$, where $\tau_1$ is the short lifetime for free NAD(P)H and bound FAD, $\tau_2$ is the long lifetime of bound NAD(P)H and free FAD, and $\alpha_1$ and $\alpha_2$ represent relative contributions from free and protein-bound NAD(P)H, respectively, and the converse for FAD) of NAD(P)H and FAD were calculated and autofluorescence imaging endpoints were averaged for all pixels within a cell cytoplasm using RStudio v. 1.2.1335 (*Team R, 2015*). For the in vivo macrophages, a custom CellProfiler pipeline segmented the macrophage cell area. Briefly, the pipeline rescaled the mCherry intensity images to be between 0 and 1 by dividing by the brightest pixel value in the image. Background was excluded by manually setting a threshold (0.15). Cells were identified using CellProfiler's default object identification. Then, each cell was manually checked and edited as

necessary to exclude background fluorescence and to include all pixels of each macrophage. Images of the optical redox ratio (intensity of NAD(P)H divided by the sum of the intensity of NAD(P)H and the intensity of FAD) and the mean fluorescence lifetime ($\tau_m = \alpha_1 \tau_1 + \alpha_2 \tau_2$; defined above) of NAD(P)H and FAD were calculated and autofluorescence imaging endpoints were averaged for all pixels within a cell using MATLAB v.9.7.01296695 (R2019b; Mathworks, Natick, MA). OMI index (*Walsh et al., 2014*; *Walsh and Skala, 2015*), the linear combination of mean-centered optical redox ratio, NAD(P)H $\tau_m$, and FAD $\tau_m$ (coefficients of 1, 1, and –1) was calculated for each cell.

## LC–MS based metabolite analysis

To analyze intracellular metabolites, metabolites were extracted from 80 to 100 cut tails with cold (on dry ice) LC–MS grade 80/20 methanol/$H_2O$ (v/v). Samples were dried under nitrogen flow and subsequently dissolved in LC–MS grade water for LC–MS analysis methods. Protein pellets were removed by centrifugation. Samples were analyzed using a Thermo Q-Exactive mass spectrometer coupled to a Vanquish Horizon ultra-high performance liquid chromatograph. Metabolites were separated on a C18 (details below) at a 0.2 ml per min flow rate and 30°C column temperature. Data was collected on full scan mode at a resolution of 70 K. Samples were loaded in water and separated on a 2.1 × 100 mm, 1.7 µM Acquity UPLC BEH C18 Column (Waters) with a gradient of solvent A (97/3 H2O/ methanol, 10 mM TBA, 9 mM acetate, pH 8.2) and solvent B (100% methanol). The gradient was: 0 min, 5% B; 2.5 min, 5% B; 17 min, 95% B; 21 min, 95% B; 21.5 min, 5% B. Data were collected on a full scan negative mode. The identification of metabolites reported was based on exact m/z and retention times, which were determined with chemical standards. Data were analyzed with Maven. Relative metabolite levels were normalized to protein content.

## Statistical analyses

Biological repeats are defined as separate clutches of embryos collected on separate days. Statistical significance was set to 0.05. Statistical analyses of autofluorescence imaging data were performed using R v.3.6.2 (https://www.R-project.org) (*R Core Team, 2019*). General linear models with gaussian errors were fit to data, where every data point represented a macrophage. For zebrafish experiments, models included *day* (biological repeat) as a blocking factor and indicators for experimental treatments or conditions. An interaction was included in models where more than one experimental factor was present (e.g. time and treatment), to determine whether effects associated with one experimental factor modified the other. All models utilized cluster-robust standard errors to account for multiple macrophages being measured within the same larvae. Model assumptions (normality of errors and constant error variance) were assessed by checking residuals with normal quantile/percentile plots and inspecting residuals versus fitted values for constant variance. Log transformation was applied to certain lifetime endpoints when these model checks revealed an issue. It is indicated in the figure legend if log transformation was applied. No adjustment for multiplicity was done. Graphical displays were generated in Python using the open-source graphing package Matplotlib (https://matplotlib.org/). Each data point in the graphical display represents a macrophage and the data for each condition is presented as a composite dotplot and boxplot; each data point is a different macrophage; each biological repeat is displayed by a different color in the dotplot; boxplots show median (central line), first, and third quartiles (lower and upper lines), and the Tukey method was employed to create the whiskers (the farthest data points that are no further than 1.5 times the interquartile range); data points beyond whiskers are considered outliers. Statistical conclusions (p values) apply to the *overall effects* when comparing the groups and also account for the cluster-correlated structure of data as described above; this is denoted by italicizing the p values. Estimated means with 95% CI and comparison between groups showing the *overall effect* computed as fold change (ratio) or simple difference with 95% CI and p values are provided separately in supplemental source data files. Proportion of TNFα+ macrophages and total macrophages at the wound, total number of macrophages in whole larvae, and tissue regrowth area to measure wound healing in zebrafish larvae were analyzed in SAS/ STAT 9.4 (SAS Institute Inc, Cary, NC) using a linear mixed model with treatment or genotype as the experimental effect, and replicate day as the random effect, with an interaction included when time was a second experimental factor. Normality and residuals were assessed and no transformations were required. Tukey adjustment for pairwise comparisons was utilized when more than two treatments were compared (i.e. wild-type, heterozygote, and homozyote mutant genotypes). Data were

graphed using Prism (GraphPad Software, Inc, San Diego, CA); each data point represents a larva and arithmetic mean with 95% CI is shown in the graphical displays.

## Code availability

All codes used for image and statistical analyses are deposited at https://github.com/skalalab/zebrafish_flim (copy archived at swh:1:rev:808637b36531d833e61c9495cb3aac4ae3723df0; *Miskolci, 2022*).

## Acknowledgements

We thank members of the Huttenlocher and Skala laboratories, notably Elizabeth S Berge, Steve Trier, Kayvan Samimi, Peter R Rehani, Emmanuel Contreras Guzman, Tiffany M Heaster and Amani Gillette, and our collaborators from the Laboratory for Optical and Computational Instrumentation (LOCI), Jayne M Squirrell and Kevin W Eliceiri, for technical assistance and valuable discussions.

## Additional information

### Funding

| Funder | Grant reference number | Author |
|---|---|---|
| National Institutes of Health | R35 GM118027 | Anna Huttenlocher |
| National Institutes of Health | R01 CA205101 | Melissa C Skala |
| National Institutes of Health | K99 GM138699 | Veronika Miskolci |
| American Heart Association | 17POST33410970 | Veronika Miskolci |
| National Institutes of Health | R21 AI159312 | Anna Huttenlocher |

The funders had no role in study design, data collection and interpretation, or the decision to submit the work for publication.

### Author contributions

Veronika Miskolci, Conceptualization, Data curation, Formal analysis, Investigation, Methodology, Visualization, Writing – original draft; Kelsey E Tweed, Data curation, Formal analysis, Investigation, Software; Michael R Lasarev, Landon J Zimmerman, Formal analysis; Emily C Britt, Formal analysis, Investigation; Alex J Walsh, Formal analysis, Investigation, Methodology; Courtney E McDougal, Mark R Cronan, Resources; Jing Fan, John-Demian Sauer, Methodology, Resources; Melissa C Skala, Conceptualization, Funding acquisition, Methodology, Resources, Supervision, Writing – review and editing; Anna Huttenlocher, Conceptualization, Funding acquisition, Methodology, Project administration, Resources, Supervision, Writing – original draft

### Author ORCIDs

Veronika Miskolci  http://orcid.org/0000-0001-7900-4626
Michael R Lasarev  http://orcid.org/0000-0002-1896-2705
Alex J Walsh  http://orcid.org/0000-0003-3832-8207
John-Demian Sauer  http://orcid.org/0000-0001-9367-794X
Anna Huttenlocher  http://orcid.org/0000-0001-7940-6254

### Ethics

Animal care and use was approved by the Institutional Animal Care and Use Committee of University of Wisconsin and strictly followed guidelines set by the federal Health Research Extension Act and the Public Health Service Policy on the Humane Care and Use of Laboratory Animal, administered by

the National Institute of Health Office of Laboratory Animal Welfare. All protocols using zebrafish and mouse in this study have been approved by the University of Wisconsin-Madison Research Animals Resource Center (protocols M005405-A02/zebrafish, M005916/mouse).

### Decision letter and Author response
Decision letter https://doi.org/10.7554/eLife.66080.sa1
Author response https://doi.org/10.7554/eLife.66080.sa2

---

## Additional files

### Supplementary files
• Transparent reporting form

### Data availability
Source data files containing numerical data used to generate the graphical displays are provided for all figures.

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
