## [Editor Report]

Immunometabolism is an emerging field, and to understand immune cell metabolism during inflammation and infection is of great interest. In this report, cutting edge microscopy techniques and innovative zebrafish models are used to characterize the metabolism of macrophages in situ. In the future, fluorescence microscopy approaches pioneered using zebrafish may illuminate strategies to therapeutically manipulate metabolism in human immune cells.

---

## [Decision Letter]

**Decision letter after peer review:**

Thank you for submitting your article "In situ fluorescence lifetime imaging of macrophage intracellular metabolism during wound repair in zebrafish" for consideration by *eLife*. Your article has been reviewed by 3 peer reviewers, one of whom is a member of our Board of Reviewing Editors, and the evaluation has been overseen by Didier Stainier as the Senior Editor. The following individual involved in review of your submission has agreed to reveal their identity: Robert Knight (Reviewer #3).

Essential revisions:

1) The authors use label free imaging methods to measure key components of immune cell metabolism. FLIM data presented appears promising, but alternative methods to support results / interpretations is lacking throughout this study.

2) Various key results appear unexpected / contradictory (for example macrophage results obtained in vitro versus in vivo (Figure 1 vs Figure 4); metabolomic monitoring of macrophages alone versus entire tissue (Figure 5))? To promote confidence in your new approach, support contradictory results with new data, alternative approaches, and/or more elaborate justification.

3) Where highlighted by the reviewers, please explain / justify technical and experimental details in depth.

*Reviewer #1 (Recommendations for the authors):*

I am enthusiastic about the imaging approach and its potential, but not yet convinced fluorescence lifetime imaging of macrophages in situ is reliable.

1. It is crucial to better understand why results in vitro do not mimic results in vivo. In the case of *Listeria* infection, controls can be performed. For example:

– Confirm that *Listeria* is intracellular during fluorescence lifetime imaging of macrophages (in vitro and in vivo).

– *Listeria* is well known to invade epithelial cells. Could this influence results described in vivo?

– If to perform fluorescence lifetime imaging of *Listeria* infected HeLa cells, would these results mimic BMDM infection in vitro or infection in vivo?

2. Metabolomics of whole tail fin tissue (not macrophages alone)

It is surprising that mass spec of whole tail fin tissue gives similar result as for imaging of single macrophages. Considering these results suggest a whole tissue response (dominated by non-macrophages), what is the significance of monitoring macrophages only (if they behave the same as the rest of the tissue)?

3. Measure / manipulate classic markers of inflammation, eg mROS and cytokines (TNF), in parallel to fluorescence lifetime measurements to help build confidence in these new results.

4. Are fluorescence lifetime measurements for non-motile cells directly comparable to those from motile cells?

5. Line 162. 'We found that GFP is suitable to image in conjunction with NAD(P)H, but it excludes the acquisition of FAD as they have overlapping spectra. However, we found mCherry to be compatible for simultaneous imaging with NAD(P)H and FAD. We also optimized imaging on live larvae.'

-Please show these data for readers to follow how conclusions are made

6. Lines 227-229. In the previous work Miskolci et al. 2019, macrophage recruitment, TNFa+ cells and *Listeria*-infected cells peak between 72 and 96hpw. However in the present work, authors only analysed 48hpw. Why?

7. Figures

– For all images: Additionally provide inset image to zoom in on a single cell to better illustrate changes with this new technique.

– In the case of Figure 1, metabolomic changes of bacteria were excluded – why is this not done for all figures involving infection?

– Statistical information can be consistently presented across all figures (eg Figure 1 directly on plots; Figure 2 presented in Table).

8. Figures 3,4,5: Understandably the authors have counted cells in the same number of embryos. However, would it be more informative to count similar number of cells for the different conditions?

*Reviewer #3 (Recommendations for the authors):*

The approach used is novel and could potentially be helpful for researchers using zebrafish to measure cell metabolism. My major criticism is that the interpretation of the intensity and fluorescence lifetime data is simplistic and does not account for the complexity of NADH/NADPH bioavailability in cells. It is also very difficult to relate such measures to cellular metabolism without some additional method for confirming this.

The paper was somewhat hard to read, with technical details insufficiently explained and results only cursorily described or described using indirect language without specifically saying what was observed (e.g. a similar trend was observed as seen previously…). The manuscript would have benefited from much clearer descriptions of the actual data obtained and brief but to the point interpretations of these results.

On a technical note, I was not quite clear about how long it took to acquire sufficient photon counts from the zebrafish larvae – I note in the methods (pg. 17, line 562) that images acquired were 256 x 256 pixels and acquisition speed was 4.6 µs per pixel with 256 time bins. As macrophages are relatively motile I wondered how precise this measure of fluorescence lifetime was for each macrophage in these experiments and would like to have a clearer description of the acquisition process in the text to clarify this point.

I was also quite surprised that there was no mention of other methods used to measure NADH/ NADPH activity in zebrafish such as the iNap biosensor described by Tao et al. (doi: 10.1038/nmeth.4306). How do the measures of NADH/NADPH by intensity measures compare to such biosensors? It would be important to know how comparable FLIM measures of NAD(P)H are to such a biosensor to enable other researchers to make an informed choice as to whether FLIM is sensitive enough to resolve biologically meaningful differences in NAD(P)H bioavailability.

Specific comments are as follows:

pg. 2: the abstract is rather opaque and would benefit from some rewriting to clarify certain statements e.g. line 46 'FLIM also resolved temporal changes in the optical redox ratio and lifetime variables of NAD(P)H..'. There is no explanation of what these lifetime variables are making this sentence hard to understand.

pg. 5 (line 143): it would be helpful to clarify that initially the authors have obtained an intensity based measure of oxidation/ reduction to measure the optical redox ratio and they also obtain lifetime measures for the different molecules. The introduction implies they are only aiming to measure lifetime so it is not obvious this is not lifetime that is initially described.

pg. 5 (line 148) and pg. 26 (line 853): When describing calculations for the optical redox ratio in mouse macrophage infected by *Listeria* the authors appear to have adjusted this relative to expression of mCherry in bacteria: 'mCherry expression in bacteria was used to subtract from lifetime images in order to exclude metabolic changes in the pathogen from that of macrophages.' I assume this means they used the mCherry as a mask to remove the lifetime signals from bacteria expressing mCherry but it hard to understand and needs to be clarified and explained.

pg. 5 (line 151): I could not find a description for FLIM measures for the in vitro macrophage culture experiments and noticed the authors only briefly commented on the changes to fluorescence lifetime that were observed. The emphasis of the paper is that FLIM is a powerful method to identify changes to the redox state of the cell so I would consider it important to address this in the text and describe what the changes were.

pg. 5 (line 163): the authors cite unpublished data that argues mCherry emissions are compatible with FLIM imaging of NAD(P)H and FAD+ but do not show any evidence for this. This statement should be justified.

pg. 6 (line 175): Calculations to measure changes to NAD(P)H fluorescence lifetime are performed by using an average of the short and long fluorescence lifetime signals NAD(P)H. This does not account for the known differences of the long lifetime (tau2) component of NADH when it is bound to different proteins. It would be helpful to know how the authors have accounted for this potential source of error when calculating the proposed FLIM redox ratio using a mean fluorescence lifetime value of NAD(P)H.

pg. 6 (line 177): how many macrophages were measured in total to obtain the relative alpha and tau values for fluorescence lifetime shown in figure 2? And were the same macrophages measured at multiple times or are the datapoint shown in Figure 2 representative of different macrophages? These results should be more fully described to allow the reader to understand how much sampling was performed when measuring FLIM.

pg. 7 (line 223): when measuring FAD+ and NAD(P)H fluorescence lifetime the authors describe a difference in the relative proportion of signals in the alpha1 fractional component of free NAD(P)H between infected and uninfected animals (shown in Figure 4) compared to the alpha1 fractional component measured when measuring NAD(P)H alone in animals expressing GFP (shown in Figure 3). It is difficult to discriminate this difference when comparing Figure 3 and 4 supplements as the significance of the difference between control and infected animals for the NAD(P)H alpha1 fraction is only shown in Figure 4 supplement. Comparable data should be shown for both sets of data to allow a comparison of the alpha1 component between them.

pg. 8 (line 250): it is difficult to discern from the text which measures show significant differences in NAD(P)H and FAD+ between burn compared to transection injury models. It would be helpful in this instance to state whether the lifetime measures are increased or decreased and how these differences reflect the redox state. As an example, the authors state that 'The trends for the differences in the mean lifetime and individual lifetime components of NAD(P)H and FAD between the burn wound and simple transection at 24 hpw were comparable to the observed differences between the infected and simple transection (Figure 5C, D, Figure S5A-F).' This does not provide the reader with detail as to what is actually different and should clearly state what such differences are.

pg. 8 (line 254): The authors state that there is a significant difference for the NAD(P)H lifetime in macrophages in the tail of burn compared to transection injury models. This is calculated from the relative contributions of tau1 and tau2 to the mean fluorescence lifetime of NAD(P)H (Figure 5 Supplement). What was not discussed is that tau1 for NAD(P)H clearly shows a difference between sterile injuries (306, 95% CI: 289-325) and burn injuries (274, 95% CI: 264-284) at 24 hpi but tau2 is almost identical between these two models. Is this change to the tau1 component important for understanding differences in the mean NAD(P)H lifetime between the two models at 24 hpi and not the tau2 component?

pg. 9 (line 303): the authors describe changes to NAD(P)H lifetimes in animals treated with Metformin and state 'the changes in NAD(P)H tau1 and tau2 also trended as expected'. It is not clear what should be expected and how these trended. The authors should clarify this statement to state what the results actually showed.

pg. 19 (line 624): the authors state that a general linear model was used to test for significant changes. How was the data fitted to the models, what were the models used, with which assumptions were the models generated and where are the results from the models? I would expect this to be made more explicit so this work could be examined critically, but the level of detail of the statistical approaches used is not sufficient to enable this and should be addressed.

pgs. 36, 38, 40.

Supplemental figures 2, 3, 4: what do the R1, R2 etc denote?

Data availability and statistical methodology:

It was not possible to comment on the statistical methods used as there was insufficient detail provided in the manuscript. It was also not possible to discern the exact details for how the acquisition of FLIM data was performed including duration of scanning for each cell and whether a cell was examined more than once.

[Editors' note: further revisions were suggested prior to acceptance, as described below.]

Thank you for resubmitting your work entitled "in vivo fluorescence lifetime imaging of macrophage intracellular metabolism during wound responses in zebrafish" for further consideration by *eLife*. Your revised article has been reviewed by 3 peer reviewers, one of whom is a member of our Board of Reviewing Editors, and the evaluation has been overseen by Didier Stainier as the Senior Editor.

The manuscript has been improved but there are some remaining issues that need to be addressed, as outlined below. In particular, 2 issues raised by the Reviewers which require further revision of the text/analysis of existing data are:

1)The authors provided a technical information in the rebuttal on how images were segmented in the TNFa expression data (previous Figure 3 current Figure 2). However I am not sure they answered my question. My point was that in the fluorescence images they show dotted outlines of macrophages which are supposed to be derived from segmentation based on cherry intensity. However, the cherry intensity within those outlines is not uniformly higher than outside the outlines – there are dark/low-intensity areas within the outlines. Conversely there are areas of high-intensity which are not included in the outlines. So it still puzzles me how these outlines are deduced because if segmentation is threshold-based the areas inside the outlines should all have intensities above a certain common threshold. Perhaps the authors need to change the brightness/contrast of the image and/or show us how they threshold the image, to validate how these cellular outlines are deduced.

2)The authors need to acknowledge that they are looking at both NADPH/NADP AND also NADH/NAD with the methods they have used. These two redox molecule pairs have quite different functions in the cytoplasm and mitochondria so it is important to recognise this point. Saying that, their methodology for discriminating oxidative vs reduced status in a cell is OK despite this but it makes it difficult to be specific about which metabolic process is likely to be occurring based on these measures. Perhaps they could update the manuscript to acknowledge/ discuss.

*Reviewer #1 (Recommendations for the authors):*

The authors have delivered a significantly revised manuscript in response to reviewer comments, where new results strongly support the use of fluorescence lifetime imaging to characterise macrophage metabolism in situ. Many interesting future directions emerge from this innovative work.

I am highly positive about this revised manuscript.

*Reviewer #2 (Recommendations for the authors):*

The authors have performed a substantial revision in response to the reviews. In response to my comments they:

– Added data that indicate the biological significance of the optical redox measurements. Using STAT6 depletion or Metformin treatment they shifted the macrophage differentiation status towards M1 or M2 and correlated this with changes in optical redox ratio measurements. The treatments led to changes in tissue repair process, indicating that the optical redox ratio measurements may have predictive value in regeneration outcome after different treatments.

– Added new references to explain limitations in choice of fluorescent reporters that can be used for this method.

– Added a comment in the paper regarding discrepancies between in vitro and in vivo data. Given this was raised by two reviewers, they could provide a stronger commentary, for example by indicating other relevant examples in the literature where in vitro and in vivo observations on cell phenotypes don’t match.

– Provided potential reasons in the rebuttal for the low magnitude of the metabolic changes observed in vitro with their system. These reasons could be also included in the manuscript in brief form.

– Provided a technical information in the rebuttal on how images were segmented in the TNFa expression data (previous Figure 3 current Figure 2). Here I am not sure they answered my question. My point was that in the fluorescence images they show dotted outlines of macrophages which are supposed to be derived from segmentation based on cherry intensity. However, the cherry intensity within those outlines is not uniformly higher than outside the outlines – there are dark/low-intensity areas within the outlines. Conversely there are areas of high-intensity which are not included in the outlines. So it still puzzles me how these outlines are deduced because if segmentation is threshold-based the areas inside the outlines should all have intensities above a certain common threshold. Perhaps the authors need to change the brightness/contrast of the image and/or show us how they threshold the image, to explain more clearly how these cellular outlines are deduced.

– Added a new Table 1 with better explanation of the interpretations of the metrics.

-explained choice of time point for measurements.

– Improved the abstract.

– Improved the discussion. I still found paragraph in lines 349-368 difficult to follow. I am not really sure what the reader should take from it. This paragraph should probably move to supplementary information, as the role seems to be to provide more technical information for some specialists.

Altogether they have generally addressed my concerns but I would just recommend some straightforward edits as indicated in the specific points above.

*Reviewer #3 (Recommendations for the authors):*

I thought the authors have addressed most of the points raised well and thoughtfully. The manuscript is much more readable and results are now clear. I think this is a valuable and helpful addition to the field and recommend publication.

I have a few additional comments/ questions that I think would strengthen the manuscript further:

1. I wondered about the declaration on line 277 that 'The modest changes in NAD(P)H lifetime endpoints are likely attributed to the modest decrease in TNFα expression in macrophages upon Metformin treatment'.

This implies that TNF-a regulates the metabolic status of the cell. Is this intended? It would be helpful to therefore clarify the potential relationship between NAD(P)H t1 and t2 and TNF-a expression.

2. how does their study correspond to the descriptions of the metabolic profiling of macrophages during regeneration described by Ratnayake et al., 2019? In this work the authors describe a role for the NAD+ regenerating enzyme Nampt in promoting regeneration through regulating NAD+ replenishment in macrophages. By measuring the NADH/ NAD+ ratios by FLIM they argue the pro-regenerative macrophages display a glycolytic phenotype. It would be helpful to compare results from this work to this study.

3. The major thrust of this work is that it is appropriate to use the ratio of NAD(P)H and FAD to approximate the relative oxidative state of a cell. The two methods used to measure NAD(P)H – intensity measures of autofluorescence and relative changes to fluorescence lifetime – do discriminate between NAD(P)H and NAD(P). However the emission wavelength of the equally important redox molecule NADH is identical to NAD(P)H so any intensity measure will include both. It is important to consider this caveat as changes to NADH levels will affect any such intensity based measures. I wondered therefore if the changes to the optical ratio in various manipulations might not always be correlated to changes of NAD(P)H tau1 and NAD(P)H tau2 due to this confounding factor? It would be greatly helpful to introduce the importance of NADH/ NAD in metabolic signalling and discuss whether the optical ratio utilised in this study would also provide an insight to changes of these critical molecules.

4. I commented on the caveat that the lifetime for bound NAD(P)H can change depending on the protein it is bound to. The authors have made the data for all measured lifetimes available and discussed it in the Discussion section (line 377), but after reading this I was left wondering what the reader could definitively determine from this measure. The last two sentences of this section in the Discussion are intriguing as they imply changes to tau1 and tau2 could be used to infer changes to different metabolic processes. Is it possible to present this with clearer predictions for what changes to tau1 and tau2 might mean in relation to the metabolism of the cells?

---

## [Author Response]

Reviewer #1 (Recommendations for the authors):I am enthusiastic about the imaging approach and its potential, but not yet convinced fluorescence lifetime imaging of macrophages in situ is reliable.

We include new data (Figure 6) and analysis that provide more convincing evidence that FLIM is a reliable way to image macrophages in situ. 1. It is crucial to better understand why results in vitro do not mimic results in vivo. In the case of Listeria infection, controls can be performed. For example:

We agree with the reviewer that is an unexpected observation. Our approach in the revision was to provide even more convincing evidence that our in situ results accurately report metabolic activity of macrophages. We have moved the in vitro results (which displayed a minor phenotype) to the supplemental data (Figure S3I-K). We include robust controls of the in vivo analysis where we include 2-DG, metformin and depletion of STAT6 that impact metabolism and are consistent with our findings with TNFa+ macrophages at the infected tail wound (Figure 2, 3) and burn wound (Figure 4). We also include new functional data on the impact of these changes on TNFa+ macrophages and regeneration of the fin. Since our submission, there is new publication (PMID: 34518542) that support our findings in zebrafish larvae.

– Confirm that Listeria is intracellular during fluorescence lifetime imaging of macrophages (in vitro and in vivo).

We provide confirmation that *Listeria* is intracellular during in vitro infection. For the in vitro experiments *Listeria* was labeled with mCherry, see Figure S3I. We also mention in the text that the mCherry label allowed us to subtract *Listeria* from the images during image analysis, so that we measure and report macrophage metabolism only. For in vivo *Listeria* infection, we also have evidence that *Listeria* is intracellular in zebrafish larvae; please see Figure 3—figure supplement 2B in our previous publication (PMID: 31259685) showing that *Listeria* co-localize with macrophages in an infected wound. We also include image set in Figure S2E of this current manuscript. Regarding labeling *Listeria* for the in vivo FLIM experiments, we were limited by the colors available for imaging and it was not feasible. There are 3 PMTs available for detection in this system, so at most we can only detect in 3 channels; one for macrophages, one for FAD and one for NAD(P)H. For the in vivo experiments we needed a label for macrophages so that we could segment them from the field of view during data analysis (this is not an issue for in vitro experiments).

– Listeria is well known to invade epithelial cells. Could this influence results described in vivo?

Yes, it is a possible reason why in vitro results are different from in vivo. in vitro measurements where macrophages are in isolation do not take into the account the input from multiplex soluble signals and other interacting cells in the microenvironment, which in vivo measurements do. This underscores the reason why we set out to test FLIM as a tool to study single cell-based intracellular metabolism in vivo, to be able to address these fundamental differences between in vitro and in vivo settings. Therefore, in the revision we improved the analysis and treatments to detect how changes in metabolism alter macrophage FLIM in tissues. We think our findings are robust and not surprisingly suggest that metabolism of macrophages is different in vitro and in vivo.

– If to perform fluorescence lifetime imaging of Listeria infected HeLa cells, would these results mimic BMDM infection in vitro or infection in vivo?

That is an interesting biological question, but this does not address the in vitro versus in vivo problem and is beyond the scope of the current paper.

2. Metabolomics of whole tail fin tissue (not macrophages alone)It is surprising that mass spec of whole tail fin tissue gives similar result as for imaging of single macrophages. Considering these results suggest a whole tissue response (dominated by non-macrophages), what is the significance of monitoring macrophages only (if they behave the same as the rest of the tissue)?

We were also surprised that the tail fin tissue metabolomics was consistent with imaging single macrophages. However, we think this finding is informative and could not have been discovered without the single cell in vivo analysis. In addition, mass spec analysis destroys the samples and is not compatible with obtaining temporal and spatial information in the same animal. Using FLIM, we can measure live samples, in situ, and obtain temporal and spatial information. Furthermore, FLIM also provides single cell-based information, which can then be used to analyze heterogeneity in the measured population. The Skala lab is developing methods to analyze population heterogeneity (PMID: 31078067, PMID: 32719514, PMID: 31737571), and we plan to apply these tools in future studies.

3. Measure / manipulate classic markers of inflammation, eg mROS and cytokines (TNF), in parallel to fluorescence lifetime measurements to help build confidence in these new results.

We imaged TNF reporter and NAD(P)H lifetime in the same cells (Figure 2A). We also treated zebrafish with Metformin to reduce the proportion of TNFa+ macrophages at the burn wound (Figure 5). Metformin is known to promote M2 polarization, and we showed previously that it reduces the number of TNFa+ macrophages (PMID: 30572006). As expected, we observed an *increase* in redox ratio in the Metformin treated larvae. We also observed improved tissue repair (Figure 6G, H). To further validate the correlation between macrophage population and FLIM endpoints, we did the opposite experiment, where we attempted to *increase* the proportion of TNFa+ macrophages. We used the recently published Stat6 zebrafish mutant (PMID: 33761328). Stat6 is required for M2 macrophage polarization (PMID: 27813830), so we expected that deletion of Stat6 would result in an increase in the proportion of TNFa+ macrophages at the burn wound. We confirmed this in revised Figure 6A, B. Based on the *increase* in TNFa+ macrophages, we predicted that wound healing would be worse compared to wild-type, and indeed we found a decrease in wound healing (revised Figure 6I, J). We also observed a decrease in redox ratio and mean lifetime of NAD(P)H (Figure 6D, F).

4. Are fluorescence lifetime measurements for non-motile cells directly comparable to those from motile cells?

This is also an interesting biological question. However, we did not test it. M1 and M2 macrophage have different metabolism and different motility characteristics (PMID: 28458726), M1 macrophages are less motile compared to M2 cells. Therefore, we expect that non-motile cells would have different intracellular metabolism compared to motile cells.

5. Line 162. 'We found that GFP is suitable to image in conjunction with NAD(P)H, but it excludes the acquisition of FAD as they have overlapping spectra. However, we found mCherry to be compatible for simultaneous imaging with NAD(P)H and FAD. We also optimized imaging on live larvae.'-Please show these data for readers to follow how conclusions are made.

Overlap between GFP and FAD is well documented in the literature, see review (PMID: 32406215), as is separation between NAD(P)H and GFP (PMID: 34321477). Our prior published studies and those of other groups have also shown that mCherry fluorescence can be separated from NAD(P)H and FAD autofluorescence (PMID: 33959597; PMID: 32509583). We have added these citations so that these methods can be replicated.

6. Lines 227-229. In the previous work Miskolci et al. 2019, macrophage recruitment, TNFa+ cells and Listeria-infected cells peak between 72 and 96hpw. However in the present work, authors only analysed 48hpw. Why?

We picked this time point based on the proportion of TNFa+ macrophages at the infected wound. We wanted to pick a time point where the macrophage populations as measured by TNFa expression would be maximally different between simple transection and the infected wound. At the infected wound, the number of macrophages peak around 72-96 hpw, however the proportion of TNFa+ macrophages stays relatively similar past 48 hpw, ~75-80%. Based on this, we selected 48 hpw. At 48 hpw, most macrophages at the simple transection wound are TNFa-, and most macrophages at the infected wound are TNFa+, so we felt this time point represented the maximal differences in macrophage populations between the 2 wound types, sufficient enough that we should detect differences by FLIM.

7. Figures– For all images: Additionally provide inset image to zoom in on a single cell to better illustrate changes with this new technique.

We opted not to include zoomed insets as FLIM image resolution is low (256x256), and the zoomed-in images would be pixelated.

– In the case of Figure 1, metabolomic changes of bacteria were excluded – why is this not done for all figures involving infection?

See answer to question 1; we cannot image 4 channels/PMTs and also image FAD and NAD(P)H. This is why we used a sterile burn wound without infection. However, given that macrophages at the burn wound had reduced redox ratio and the endpoints of NAD(P)H trended similarly as at the infected wound (see Table 2), the burn wound provides a simpler inflammatory damage model for FLIM.

– Statistical information can be consistently presented across all figures (eg Figure 1 direclty on plots; Figure 2 presented in Table).

We have revised the statistical information to address these concerns.

To reduce confusion, we removed the tables from the Figures, and created a excel file (Table 2S) summarizing the mean+/-95%CI for each endpoint, overall effects (fold change or simple difference) between groups and corresponding p values for every Figure.

8. Figures 3,4,5: Understandably the authors have counted cells in the same number of embryos. However, would it be more informative to count similar number of cells for the different conditions?

The differences in the number of cells are due to the nature of the wound; the simple transection is the least inflammatory compared to the infected or burn wounds, so that wound recruits fewer cells. We based the analysis on the advice of our statistician.

Reviewer #3 (Recommendations for the authors):The approach used is novel and could potentially be helpful for researchers using zebrafish to measure cell metabolism. My major criticism is that the interpretation of the intensity and fluorescence lifetime data is simplistic and does not account for the complexity of NADH/NADPH bioavailability in cells. It is also very difficult to relate such measures to cellular metabolism without some additional method for confirming this.

We agree with the reviewer. In this manuscript we focused on correlating the changes in the redox ratio and lifetimes with the macrophage population; this is also why we did the Metformin experiment in Figure 5, where we modulated the macrophage population and expected changes in the redox ratio and lifetimes based on the trends established at the infected and burn wounds (Table 2). But yes, we have not correlated the changes with a specific metabolic pathway; correlating changes in autofluorescence measurements with specific metabolic pathway is a current challenge and the focus of the field itself. However, the 2DG experiment in Figure 1 provides some hints; inhibiting glycolysis has the same effect on the redox ratio and NAD(P)H lifetime endpoints (t_m_, t_1_, t_2_ and a_1_) in macrophages as a pro-inflammatory M1 polarization (more TNFa+ cells), as summarized in Table 2. This suggests that M1-like (TNFa+) macrophages have reduced glycolysis. However, making correlations to specific metabolic changes will be the goal of future studies. We are also developing additional reporter lines, so we can also better characterize macrophage population and correlate them with FLIM measurements. Currently there is only M1 reporter line available (TNF reporter) for live imaging, and we are developing M2 reporter lines. We could potentially perform immunostaining for M2 markers, however we are performing the FLIM on live zebrafish.

The optical redox ratio measures the oxidation-reduction state of the cell, while the NAD(P)H and FAD mean lifetimes report on protein-binding activities, proximity to quenchers, and microenvironmental factors [PMID: 32406215]. These autofluorescence endpoints have been correlated with traditional measures of metabolism including oxygen consumption [PMID: 27300321] and metabolite levels [PMID: 24305550] but provide unique sources of contrast. Unlike traditional measurements, these autofluorescence imaging endpoints can monitor single cells over time within intact samples, which provides important context for in vivo metabolic changes. This non-destructive single-cell insight is advantageous even with the caveat of multiple sources of contrast. Future studies will continue to define these sources of contrast as the technology develops.

The paper was somewhat hard to read, with technical details insufficiently explained and results only cursorily described or described using indirect language without specifically saying what was observed (e.g. a similar trend was observed as seen previously…). The manuscript would have benefited from much clearer descriptions of the actual data obtained and brief but to the point interpretations of these results.

We have revised the manuscript as requested by the reviewer.

On a technical note, I was not quite clear about how long it took to acquire sufficient photon counts from the zebrafish larvae – I note in the methods (pg. 17, line 562) that images acquired were 256 x 256 pixels and acquisition speed was 4.6 µs per pixel with 256 time bins. As macrophages are relatively motile I wondered how precise this measure of fluorescence lifetime was for each macrophage in these experiments and would like to have a clearer description of the acquisition process in the text to clarify this point.

We have clarified this issue in the text. The duration of integration time was 60 seconds and was selected based on the minimum time needed to collect sufficient number of photons to allow for the reliable fitting of the resulting exponential decay to a two-component model in SPCImage, taking into account that the cells are live and mobile and that we wanted to minimize photobleaching. This acquisition time should not be affected by macrophage motility.

I was also quite surprised that there was no mention of other methods used to measure NADH/ NADPH activity in zebrafish such as the iNap biosensor described by Tao et al. (doi: 10.1038/nmeth.4306). How do the measures of NADH/NADPH by intensity measures compare to such biosensors? It would be important to know how comparable FLIM measures of NAD(P)H are to such a biosensor to enable other researchers to make an informed choice as to whether FLIM is sensitive enough to resolve biologically meaningful differences in NAD(P)H bioavailability.

Thank you for pointing out this biosensor. We agree with the reviewer it would be very useful to make these comparisons. We plan to do this in future, but because of technical issues it is beyond the scope of the current manuscript. We would need a BFP-tagged macrophage reporter line to be compatible with the iNap biosensor. However, we have been unable to generate such line after multiple attempts; we believe it is too dim when driven from the macrophage promoter. We are in the process of obtaining mTagBFP2, that is approximately 5 times brighter compared to its parental mTagBFP. Alternatively, far red options can be pursued, however success with BFP is more likely.

Specific comments are as follows:pg. 2: the abstract is rather opaque and would benefit from some rewriting to clarify certain statements e.g. line 46 'FLIM also resolved temporal changes in the optical redox ratio and lifetime variables of NAD(P)H..'. There is no explanation of what these lifetime variables are making this sentence hard to understand.

Lifetime variables refer to the individual endpoints we calculate and then use to derive the mean lifetime (t_m_) of the NAD(P)H and FAD; the individual endpoints are t_1_, t_2_, α_1_ and α_2_ and t_m_ = t_1α1_ + t_1_α_2_. We simplified this sentence to make it less confusing to the reader.

pg. 5 (line 143): it would be helpful to clarify that initially the authors have obtained an intensity based measure of oxidation/ reduction to measure the optical redox ratio and they also obtain lifetime measures for the different molecules. The introduction implies they are only aiming to measure lifetime so it is not obvious this is not lifetime that is initially described.

We improved the text to clarify this issue.

pg. 5 (line 148) and pg. 26 (line 853): When describing calculations for the optical redox ratio in mouse macrophage infected by Listeria the authors appear to have adjusted this relative to expression of mCherry in bacteria: 'mCherry expression in bacteria was used to subtract from lifetime images in order to exclude metabolic changes in the pathogen from that of macrophages.' I assume this means they used the mCherry as a mask to remove the lifetime signals from bacteria expressing mCherry but it hard to understand and needs to be clarified and explained.

Yes, this assumption is correct, and we clarified it in the text.

pg. 5 (line 151): I could not find a description for FLIM measures for the in vitro macrophage culture experiments and noticed the authors only briefly commented on the changes to fluorescence lifetime that were observed. The emphasis of the paper is that FLIM is a powerful method to identify changes to the redox state of the cell so I would consider it important to address this in the text and describe what the changes were.

The optical redox ratio measures the oxidation-reduction state of the cell, while lifetime primarily reflects protein-binding activity of the coenzymes. We minimized the discussion of the changes in lifetime measurements per se for the in vitro data, because we felt that the only data in this set of experiments that we could compare to the mass spec results of the cited study (Gillmaier 2012, PMID: 23285016) was the optical redox ratio. Gillmaier et al. showed that *Listeria* infection of murine bone marrow derived macrophages was associated with increased glycolytic activity. We found a slight, but significant increase in the redox ratio, which is consistent with an increase in glycolysis.

pg. 5 (line 163): the authors cite unpublished data that argues mCherry emissions are compatible with FLIM imaging of NAD(P)H and FAD+ but do not show any evidence for this. This statement should be justified.

Several prior studies have shown that mCherry fluorescence does not corrupt the emission from NAD(P)H and FAD. We have cited two of these papers for reference [PMID: 33959597; PMID: 32509583]. We provide a representative phasor plot (Author response image 1), where the mCherry lifetimes are plotted in red and are clearly separated from the NAD(P)H (blue) and FAD (green) fluorescence lifetimes.

**Author response image 1. sa2fig1:** 

pg. 6 (line 175): Calculations to measure changes to NAD(P)H fluorescence lifetime are performed by using an average of the short and long fluorescence lifetime signals NAD(P)H. This does not account for the known differences of the long lifetime (tau2) component of NADH when it is bound to different proteins. It would be helpful to know how the authors have accounted for this potential source of error when calculating the proposed FLIM redox ratio using a mean fluorescence lifetime value of NAD(P)H.

Our optical redox ratio is calculated from the intensity of NAD(P)H divided by the sum of intensities from NAD(P)H and FAD. This optical redox ratio is calculated from the sum of all photons in the lifetime decay, so it is not a FLIM redox ratio but the traditional intensity redox ratio measured in photon counting mode. The mean lifetime (τ_m_) is the weighted average of the short and long lifetimes of NAD(P)H (τ_m_ = α_1_τ_1_ + α_2_τ_2_). This mean lifetime is reported separately from the optical redox ratio, which is an intensity measurement that is not calculated from the mean lifetime of NAD(P)H. We also report individual lifetime components in the supplemental graphs for both NAD(P)H and FAD (α_1_, τ_1_, τ_2_). Therefore, all the data is available for interpretation, so that changes in the long lifetime can be evaluated separately from changes in the mean lifetime and intensity-based optical redox ratio. We have clarified this in the text and in a new Table 1.

pg. 6 (line 177): how many macrophages were measured in total to obtain the relative alpha and tau values for fluorescence lifetime shown in figure 2? And were the same macrophages measured at multiple times or are the datapoint shown in Figure 2 representative of different macrophages? These results should be more fully described to allow the reader to understand how much sampling was performed when measuring FLIM.

For each data set we report the number of cells and zebrafish larvae in the figure legends and the supplemental figures. Each data point is a different macrophage. We included a statement in the methods section that images were acquired in a single plane and then we moved to a different plane to avoid measuring the same cell twice.

pg. 7 (line 223): when measuring FAD+ and NAD(P)H fluorescence lifetime the authors describe a difference in the relative proportion of signals in the alpha1 fractional component of free NAD(P)H between infected and uninfected animals (shown in Figure 4) compared to the alpha1 fractional component measured when measuring NAD(P)H alone in animals expressing GFP (shown in Figure 3). It is difficult to discriminate this difference when comparing Figure 3 and 4 supplements as the significance of the difference between control and infected animals for the NAD(P)H alpha1 fraction is only shown in Figure 4 supplement. Comparable data should be shown for both sets of data to allow a comparison of the alpha1 component between them.

We revised the text to address this concern.

pg. 8 (line 250): it is difficult to discern from the text which measures show significant differences in NAD(P)H and FAD+ between burn compared to transection injury models. It would be helpful in this instance to state whether the lifetime measures are increased or decreased and how these differences reflect the redox state. As an example, the authors state that 'The trends for the differences in the mean lifetime and individual lifetime components of NAD(P)H and FAD between the burn wound and simple transection at 24 hpw were comparable to the observed differences between the infected and simple transection (Figure 5C, D, Figure S5A-F).' This does not provide the reader with detail as to what is actually different and should clearly state what such differences are.

We improved the text throughout to make the results more clear.

pg. 8 (line 254): The authors state that there is a significant difference for the NAD(P)H lifetime in macrophages in the tail of burn compared to transection injury models. This is calculated from the relative contributions of tau1 and tau2 to the mean fluorescence lifetime of NAD(P)H (Figure 5 Supplement). What was not discussed is that tau1 for NAD(P)H clearly shows a difference between sterile injuries (306, 95% CI: 289-325) and burn injuries (274, 95% CI: 264-284) at 24 hpi but tau2 is almost identical between these two models. Is this change to the tau1 component important for understanding differences in the mean NAD(P)H lifetime between the two models at 24 hpi and not the tau2 component?

The mean fluorescence lifetime is a weighted average of the short and long lifetime components (τ_m_ = α_1_τ_1_ + α_2_τ_2_). Therefore, data in the supplemental figures can be used to understand changes in this mean lifetime. In this case, the decrease in NAD(P)H τ_m_ for burn compared to transection wound at 24 hours is driven by a decrease in τ_1_ and an increase in α_1_. This can be interpreted as an increase in the pool of free NADH(P)H for the burn compared to transection wound at 24 hours post wound. This has been clarified in the revised text.

pg. 9 (line 303): the authors describe changes to NAD(P)H lifetimes in animals treated with Metformin and state 'the changes in NAD(P)H tau1 and tau2 also trended as expected'. It is not clear what should be expected and how these trended. The authors should clarify this statement to state what the results actually showed.

We revised the manuscript as requested by the reviewer.

pg. 19 (line 624): the authors state that a general linear model was used to test for significant changes. How was the data fitted to the models, what were the models used, with which assumptions were the models generated and where are the results from the models? I would expect this to be made more explicit so this work could be examined critically, but the level of detail of the statistical approaches used is not sufficient to enable this and should be addressed.

We provided more detail on the statistical methods.

pgs. 36, 38, 40.Supplemental figures 2, 3, 4: what do the R1, R2 etc denote?

R stands for repeat and has been clarified in the text.

Data availability and statistical methodology:It was not possible to comment on the statistical methods used as there was insufficient detail provided in the manuscript.

We provided more detail on the statistical methods in the revised manuscript.

It was also not possible to discern the exact details for how the acquisition of FLIM data was performed including duration of scanning for each cell and whether a cell was examined more than once.

We have revised the text as requested by the reviewer.

[Editors' note: further revisions were suggested prior to acceptance, as described below.]

Reviewer #2 (Recommendations for the authors):The authors have performed a substantial revision in response to the reviews. In response to my comments they:– Added a comment in the paper regarding discrepancies between in vitro and in vivo data. Given this was raised by two reviewers, they could provide a stronger commentary, for example by indicating other relevant examples in the literature where in vitro and in vivo observations on cell phenotypes don’t match.

We edited as suggested; see lines 329-333.

– Provided potential reasons in the rebuttal for the low magnitude of the metabolic changes observed in vitro with their system. These reasons could be also included in the manuscript in brief form.

We included this information in the figure legend; lines 830-840.

– Provided a technical information in the rebuttal on how images were segmented in the TNFa expression data (previous Figure 3 current Figure 2). Here I am not sure they answered my question. My point was that in the fluorescence images they show dotted outlines of macrophages which are supposed to be derived from segmentation based on cherry intensity. However, the cherry intensity within those outlines is not uniformly higher than outside the outlines – there are dark/low-intensity areas within the outlines. Conversely there are areas of high-intensity which are not included in the outlines. So it still puzzles me how these outlines are deduced because if segmentation is threshold-based the areas inside the outlines should all have intensities above a certain common threshold. Perhaps the authors need to change the brightness/contrast of the image and/or show us how they threshold the image, to explain more clearly how these cellular outlines are deduced.

We apologize for the confusion. The source of confusion is that macrophages are labeled with membrane-localized mCherry, so there won’t be a uniform signal throughout the cell, and the outlines were placed *over* the cell membrane. We adjusted the contrast and placed the outlined around the cell. In Author response image 2 we included original and brightness auto-adjusted (in Fiji) images without outlines. In the figure we outline only a few as examples, which is stated in the legend. The mCherry image in Figure 2 was replaced by this auto-adjusted version shown in Author response image 2. The corresponding GFP image was also auto-adjusted in Fiji and replaced in Figure 2.

– Added a new Table 1 with better explanation of the interpretations of the metrics.-explained choice of time point for measurements.– Improved the abstract.– Improved the discussion. I still found paragraph in lines 349-368 difficult to follow. I am not really sure what the reader should take from it. This paragraph should probably move to supplementary information, as the role seems to be to provide more technical information for some specialists.

We clarified this paragraph; lines 355-366. Our intention was to contextualize changes in the mean lifetime of NAD(P)H that were observed with M1-like (TNFα+) macrophage populations. We wanted to clarify that the NAD(P)H lifetime does not provide information about any specific pathway because NAD(P)H has over 300 binding partners. Additionally, changes in the fluorescence lifetime of NAD(P)H are multi-faceted, and can be attributed to changes in binding activity, preferred binding partners, and microenvironmental factors (e.g., pH). We hope that the edits improve clarity. Altogether they have generally addressed my concerns but I would just recommend some straightforward edits as indicated in the specific points above.Reviewer #3 (Recommendations for the authors):

I thought the authors have addressed most of the points raised well and thoughtfully. The manuscript is much more readable and results are now clear. I think this is a valuable and helpful addition to the field and recommend publication.I have a few additional comments/ questions that I think would strengthen the manuscript further:1. I wondered about the declaration on line 277 that 'The modest changes in NAD(P)H lifetime endpoints are likely attributed to the modest decrease in TNFα expression in macrophages upon Metformin treatment'.This implies that TNF-a regulates the metabolic status of the cell. Is this intended? It would be helpful to therefore clarify the potential relationship between NAD(P)H t1 and t2 and TNF-a expression.

We agree that this statement is confusing. We use TNFa expression strictly as a surrogate measure of macrophage polarization towards pro-inflammatory M1-like subsets, as TNFa is a well-established marker of M1 macrophages in humans, mice and zebrafish. What was meant here is that the *magnitude* by which we were able to modulate macrophage polarization, as measured by changes in the fraction of TNFa+ macrophages, was modest (~15% decrease by Metformin and ~20% increase by Stat6 depletion). What we meant to imply is that if we were able to cause a larger change in macrophage polarization, let’s say a decrease or increase by 50%, the differences in FLIM readouts would have been larger also. We see this correlation when we compare macrophages at the different wound models. The percentage of TNFa+ macrophages at the infected wound is ~75%, at the burn wound it’s ~50-60% and the simple transection ~10% (as characterized in Miskolci et al. *eLife* 2019). The differences in FLIM readouts were larger when comparing infected wound versus simple transection (Figure 3) than when comparing burn wound versus simple transection (Figure 4). Even though we used TNFa expression as a marker of M1 polarization, it is possible that TNFa plays a role in macrophage metabolism. This could be tested by TNFa inhibitors for instance. However, we did not intend to speculate on this possibility at this point, therefore we clarified this sentence; lines 277-279.2. How does their study correspond to the descriptions of the metabolic profiling of macrophages during regeneration described by Ratnayake et al., 2019? In this work the authors describe a role for the NAD+ regenerating enzyme Nampt in promoting regeneration through regulating NAD+ replenishment in macrophages. By measuring the NADH/ NAD+ ratios by FLIM they argue the pro-regenerative macrophages display a glycolytic phenotype. It would be helpful to compare results from this work to this study.

Thank you for mentioning this work. We included this reference in the discussion; line 399-401. We believe our conclusions that a *reduced* intracellular metabolism support regeneration (as indicated by the time-related shift toward a higher optical redox ratio, which means a more reduced intracellular redox state (Figure 4B) and our Metformin data that shows that shifting the intracellular redox state to a more reduced environment (Figure 5D) is associated with better wound healing (Figure 6H)) is consistent with the observations of Ratnayake *et al.* (PMID: 33568815), as a more reduced intracellular redox state is likely caused by an increase in glycolysis.

3. The major thrust of this work is that it is appropriate to use the ratio of NAD(P)H and FAD to approximate the relative oxidative state of a cell. The two methods used to measure NAD(P)H – intensity measures of autofluorescence and relative changes to fluorescence lifetime – do discriminate between NAD(P)H and NAD(P). However the emission wavelength of the equally important redox molecule NADH is identical to NAD(P)H so any intensity measure will include both. It is important to consider this caveat as changes to NADH levels will affect any such intensity based measures. I wondered therefore if the changes to the optical ratio in various manipulations might not always be correlated to changes of NAD(P)H tau1 and NAD(P)H tau2 due to this confounding factor? It would be greatly helpful to introduce the importance of NADH/ NAD in metabolic signalling and discuss whether the optical ratio utilised in this study would also provide an insight to changes of these critical molecules.

We included a statement in the discussion about the caveat that intensity measurements do not distinguish between NADPH and NADH due to overlapping spectra; lines 346-348.

4. I commented on the caveat that the lifetime for bound NAD(P)H can change depending on the protein it is bound to. The authors have made the data for all measured lifetimes available and discussed it in the Discussion section (line 377), but after reading this I was left wondering what the reader could definitively determine from this measure. The last two sentences of this section in the Discussion are intriguing as they imply changes to tau1 and tau2 could be used to infer changes to different metabolic processes. Is it possible to present this with clearer predictions for what changes to tau1 and tau2 might mean in relation to the metabolism of the cells?

We clarified this paragraph. Please see our response to Reviewer #2 comment “improved the discussion. I still found paragraph in lines 349-368 difficult to follow…”.